# Reliable Data Transmission in Underwater Wireless Sensor Networks Using a Cluster-Based Routing Protocol Endorsed by Member Nodes

Kaveripakam Sathish [1], Monia Hamdi [2], Ravikumar Chinthaginjala [1], Giovanni Pau [3], Amel Ksibi [4,*], Rajesh Anbazhagan [5], Mohamed Abbas [6,7] and Mohammed Usman [6]

1   School of Electronics Engineering, Vellore Institute of Technology, Vellore 632014, India
2   Department of Information Technology, College of Computer and Information Sciences, Princess Nourah bint Abdulrahman University, P.O. Box 84428, Riyadh 11671, Saudi Arabia
3   Faculty of Engineering and Architecture, Kore University of Enna, 94100 Enna, Italy
4   Department of Information Systems, College of Computer and Information Sciences, Princess Nourah bint Abdulrahman University, P.O. Box 84428, Riyadh 11671, Saudi Arabia
5   School of Electrical and Electronics Engineering, SASTRA University, Thanjavur 613401, India
6   Electrical Engineering Department, College of Engineering, King Khalid University, P.O. Box 960, Abha 61421, Saudi Arabia
7   Electronics and Communications Department, College of Engineering, Delta University for Science and Technology, Gamasa 35712, Egypt
*   Correspondence: amelksibi@pnu.edu.sa

**Abstract:** Considering Underwater Wireless Sensor Networks (UWSNs) have limited power resources (low bandwidth, long propagation delays, and non-rechargeable batteries), it is critical that they develop solutions to reduce power usage. Clustering is one solution because it not only saves energy consumption but also improves scalability and data integrity. The design of UWSNs is vital to the development of clustering algorithms. The limited energy of sensor nodes, narrow transmission bandwidth, and unpredictable topology of mobile Underwater Acoustic Wireless Sensor Networks (UAWSNs) make it challenging to build an effective and dependable underwater communication network. Despite its success in data dependability, the acoustic underwater communication channel consumes the greatest energy at a node. Recharging and replacing a submerged node's battery could be prohibitively expensive. We propose a network architecture called Member Nodes Supported Cluster-Based Routing Protocol (MNS-CBRP) to achieve consistent information transfer speeds by using the network's member nodes. As a result, we use clusters, which are produced by dividing the network's space into many minute circular sections. Following that, a Cluster Head (CH) node is chosen for each circle. Despite the fact that the source nodes are randomly spread, all of the cluster heads are linked to the circle's focal point. It is the responsibility of the MNS-CBRP source nodes to communicate the discovered information to the CH. The discovered data will then be sent to the CH that follows it, and so on, until all data packets have been transferred to the surface sinks. We tested our techniques thoroughly using QualNet Simulator to determine their viability.

**Keywords:** UWSN; member nodes; routing protocol; cluster head; clustering and routing

## 1. Introduction

The development and implementation of underwater acoustic sensor networks are critical to the realization of the Internet of Underwater Things (IoUT). This technique makes it easier to find and exploit marine resources [1,2]. The IoUT is a new component of the Internet of Things (IoT) that includes a global network of intelligent subsea devices. The IoUT will almost certainly enable a wide range of beneficial applications, including environmental monitoring and underwater exploration, as well as disaster prevention and management. When implemented in these specific ways, IoUT is regarded as a

potentially useful technology for the development of smart cities. Underwater wireless sensor networks, also known as UWSNs, have demonstrated a great deal of potential as dependable network infrastructure to support the concept of IoUT.

Due to the fact that ocean currents drive some nodes in the dynamic topology of the UAWSN to migrate farther from their origins than others, the former nodes must consume more transmission energy to reach the latter nodes. This directly contributes to the unbalanced distribution of UAWSN node energy usage. In addition, the limited transmission capacity and significant propagation latency of underwater communications always result in a relatively low utilization rate for wireless channels in UAWSNs [3]. This holds true regardless of the circumstances. Due to these qualities, an underwater transmission protocol is robust and capable of maximizing the available channels, balancing the amount of power consumed by the nodes.

Underwater communication methods, including acoustic, optical, and electromagnetic waves, make the wireless interconnection of smart sensors in aquatic environments possible. UWSN has become an essential component of a wide range of underwater applications as sensor technology, and underwater protocols have advanced. Tsunamis, oil spills, drinking water quality, territorial seas, enemy submarines, mines, coral reefs, aquatic species, underwater laboratories, and other applications are just a few of the numerous prospective applications for such technology [4]. The primary purpose of UWSN applications is to notify emergency services. For example, if the temperature or pH measurement reaches a certain threshold, the deployed UWSN will alert the offshore station. Some underwater applications are already in operation, such as the Sea web [5] surveillance underwater acoustic network in the United States and NEPTUNE [6] in Canada. Researchers have been driven to expand their studies of UWSN due to the government's intense interest in the numerous subsea uses. Furthermore, the US PLUSNet (Persistent Littoral Undersea Surveillance Network) [7] is an antisubmarine warfare underwater network maintained by the US Naval Research Office. The PLUSNet infrastructure includes sensors and robotic vehicles such as Autonomous Underwater Vehicles (AUVs) and Remotely Operated Vehicles (ROVs). Each of the aforementioned surveillance networks has a common goal: to maintain constant vigilance. Today, most surveillance systems communicate through sound waves. Furthermore, these networks are only partially automated; they include both stationary and moving sensors, as well as smart, self-driving cars. Two major problems of surveillance networks are untrustworthy data security and a lack of privacy for gathered information. Figure 1 depicts the circular illustration of the proposed architectural design.

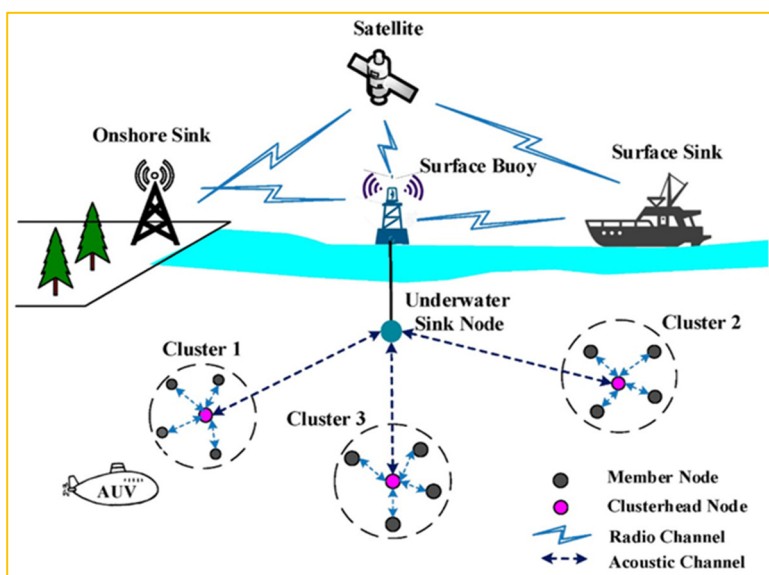

**Figure 1.** A circular illustration of the architectural design.

Because of its capacity to warn remote stations, excellent data security, and dependable data transit, our research led us to endorse the UWSN approach. Electromagnetic and radio waves have a difficult time traveling through UWSN without being severely muted due to the characteristics of water. Some hope can be retained by staying in touch below ground via radio waves operating at extremely low frequencies. However, because of the reduced bandwidth generated by this low-frequency wave, transmissions became irregular [8]. As a result, the team chose acoustic waves as the next best alternative.

The acoustic wave, often known as the sound wave, is used for the majority of underwater communication [9]. Acoustic waves can transfer high-quality data and images when paired with acoustic modems. Underwater, acoustic waves can travel up to 100 km. As one gets away from a source, the available bandwidth drops. Even in difficult underwater conditions, acoustic waves may safely carry data over long distances and at low frequencies. However, because of the softer component of the acoustic signal, an acoustic wave is not a comprehensive solution. UWSN disadvantages include slower data transmission speeds, longer communication delays, greater energy costs, and shorter network lifetimes [10].

Figure 2 represents a UWSN interaction model used in the real scenario. The clustering of optical sensors is a critical component of the proposed UWSN concept; however, it is not commonly implemented in current systems. A search of the relevant literature indicated that underwater clustering is nearly exclusively employed for acoustic sensors, while very little is written on optical sensor clustering in the deep sea. This is in contrast to the scenario on land, where much more information is available. However, because of our research into the numerous current clustering strategies for acoustic nodes, we were able to refine our own strategy for grouping optical nodes [11].

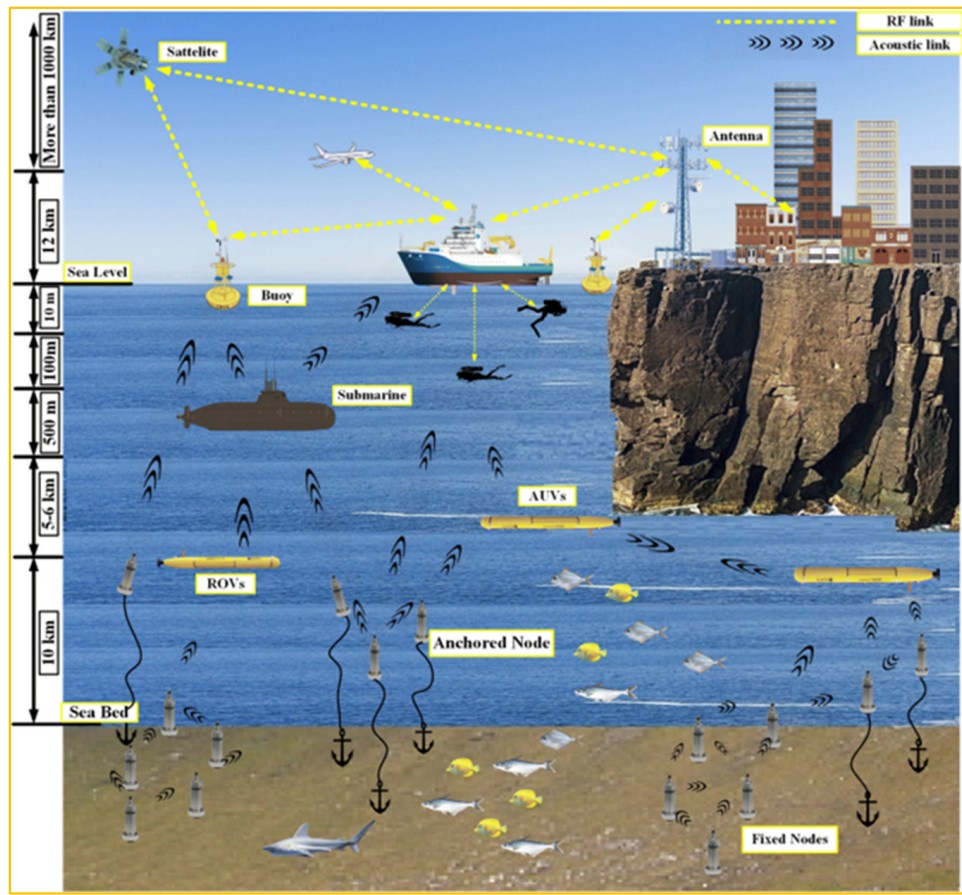

**Figure 2.** UWSN interaction model used in the real scenario.

We needed to broaden our understanding of generalized clustering schemes in UWSN; thus, we researched many acoustic sensor clustering techniques presented in the published literature. The first significant finding is that the great majority of UWSN clustering methods are just modified and better versions of Terrestrial Wireless Sensor Networks (TWSN) clustering strategies. Algorithms such as Low-Energy Adaptive Clustering Hierarchy (LEACH) [12], Group Adaptive Filtering (GAF) [13], and Hybrid Energy Efficient Distributed Clustering (HEED) [14] are adopted from their terrestrial counterparts and applied in the underwater environment, but with a few adjustments to account for the unique challenges of the marine setting. To begin, an improved, condensed, and effective clustering technique is required to deal with the significant propagation delays caused by water as well as the non-rechargeable nodes found in underwater networks. Second, the United States Navy Wireless Network will never include the Global Positioning System (GPS). Because high-frequency radio waves cannot go very far into the ocean, any unique suggestions for developing a dependable and energy-efficient clustering technique that takes into account the peculiarities of UWSN would be greatly appreciated. The authors present a clustering technique for an underwater acoustic wireless sensor network that operates on a three-dimensional grid. Following the segmentation of the entire UAWSN network into a number of grids, cluster heads are selected using a sleep/wake cycle. The network that surrounds the cluster head is built by nodes that respond to a message broadcast by the cluster head.

The TWSN GAF method [15] has been modified to function in 3D underwater acoustic sensor networks. The initial implementation of the GAF method required only a simple two-dimensional data structure. BGAF is an acronym that stands for "Based on the GAF algorithm", which also gives rise to the name. When selecting a cluster head, the quantity of unused energy and the distance between the sound sensors and the surface buoys are taken into account. When neighboring head nodes consume too much energy, the BGAF algorithm immediately activates its built-in transit mechanism.

The authors of [16] describe a K-means-based improved version of the TWSN clustering technique that may be employed for underwater acoustic sensor networks. When choosing cluster heads, the density of the nodes in the ocean as well as their depth, are taken into account. The previous technique had limitations such as excessive energy use and inconsistent clustering, but the most recent version addresses these difficulties. The previous method has flaws [17,18].

The following is a concise synopsis of the key contributions made by the paper:

- To evaluate the energy efficiency of the Source Tree Adaptive Routing-Least Overhead Routing Approach (STAR-LORA) routing protocol as the number of underwater wireless sensor nodes increases;
- To evaluate the energy trade-off between receiver and transmitter modes;
- To propose a suitable routing protocol for an underwater wireless sensor network, taking into account the desired levels of transmitted and received energy;
- To provide a protocol that would be suitable for use in an underwater wireless sensor network.

The remainder of this manuscript is organized as follows. Section 2 summarizes the Scenario of the Proposed Underwater Network. Section 3 introduces the reader Proposed MNS-CBRP design parameters. Results and discussions will be demonstrated in Section 4. This will include different deployments of applications. Section 5 presents an overview of the work.

## 2. Scenario of the Proposed Underwater Network

This section explains the idea of the scenario of the proposed underwater network. Then, we look at some more popular methods for locating underwater things. Figure 3a,b represent a UWSN architecture.

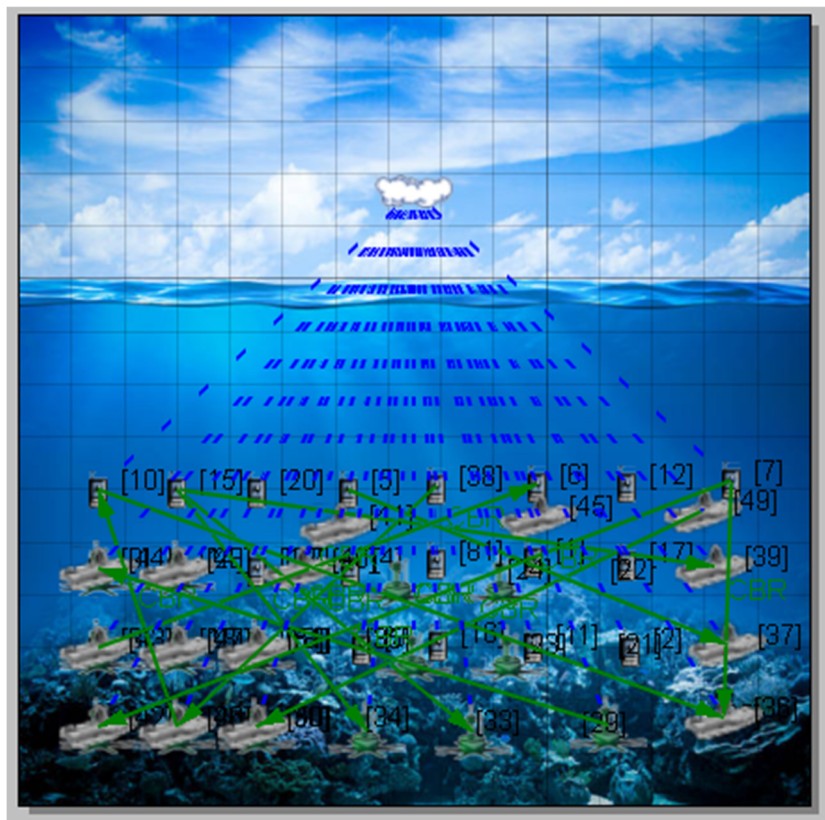

(**a**)

(**b**)

**Figure 3.** (**a**) The MNS-CBRP schemes envisioned the architecture of UWSNs. (**b**) The run view of MNS-CBRP schemes envisioned the architecture of UWSNs.



Using a Constant Bit Rate (CBR), it is possible to connect to existing networks. It is being given thought to add a Teletype network Protocol (Telnet), Supervisory–frame (S-frame), and Generic–File Transfer Protocol (Gen-FTP) into the proposed network. The QualNet simulator was built with a configuration of 1500 square meters per side and 240 nodes connecting the Telnet, S-frame, and Gen-FTP applications in order to match the predicted conditions. Only 36 nodes are sensors, four nodes are spaceships, and the remaining 200 nodes are electronic gadgets of various types. The duration of the simulation will be 3 min and 10 s in total. Random Waypoint Mobility is the employed form of node mobility, and its speeds can range between 1.2 and 3.2 m per second [19–23]. As routing protocols, only STAR-LORA, Optimized Link State Routing (OLSR), and Location-Aided Routing (LAR1) protocols were utilized initially. The simulator's built-in graphs were analyzed immediately following the conclusion of the test. The overall quantity of energy consumed during transmission and reception may then be determined, making this an essential performance indicator.

The entirety of the QualNet simulator development platform is depicted in Figure 3. The platform encourages users to develop original programs, simulate their models, and conduct statistical analyses of important performance indicators. QualNet is an IEEE 802.11-based, license-free wireless local area network. Using a drag-and-drop interface or a predetermined placement model, sensors can be positioned on the virtual landscape [24,25]. The occurrence of an event is calculated by taking the sensor's position and range into account. Figure 4 shows the MNS-CBRP process flowchart.

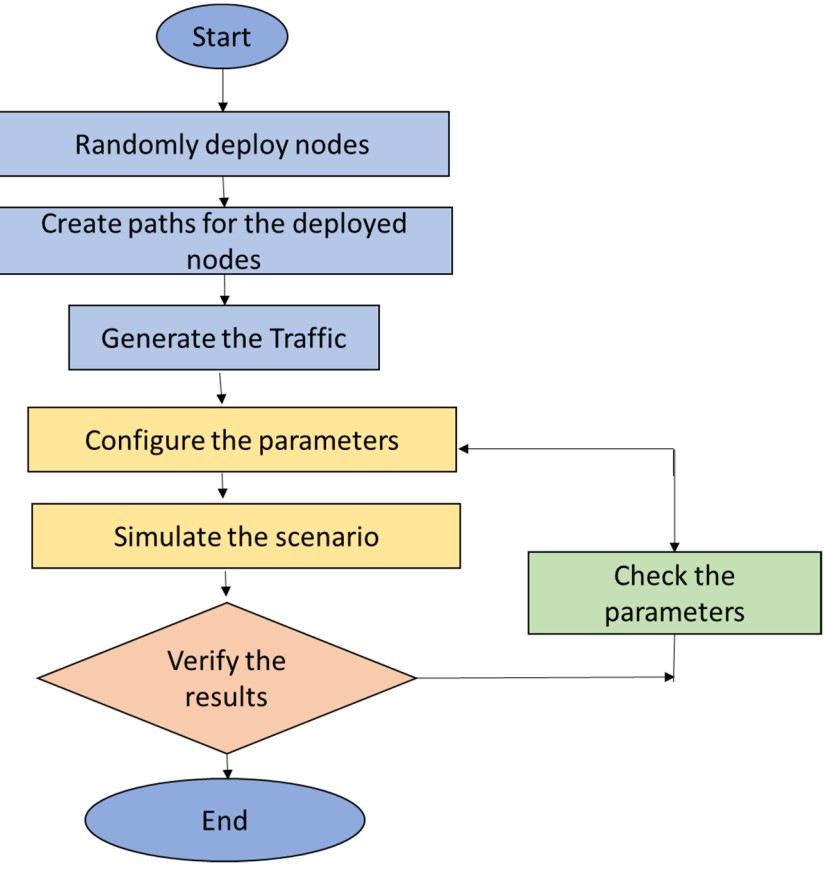

**Figure 4.** MNS-CBRP process flowchart.

## 3. Proposed MNS-CBRP Design Parameters

In the section that follows, we shall describe the cooperative UWSN model. In Figure 5, the yellow underwater acoustic sensors represent CH nodes, while the white ones represent non-CH nodes, indicating the structure of the clustering. The black line represents

direct communication between two nodes in the CH network, whereas the blue line represents communication between a CH node and the Base Station (BS). Dynamic Coded Collaboration (DCC) is shown by the red dashed line [26–28].

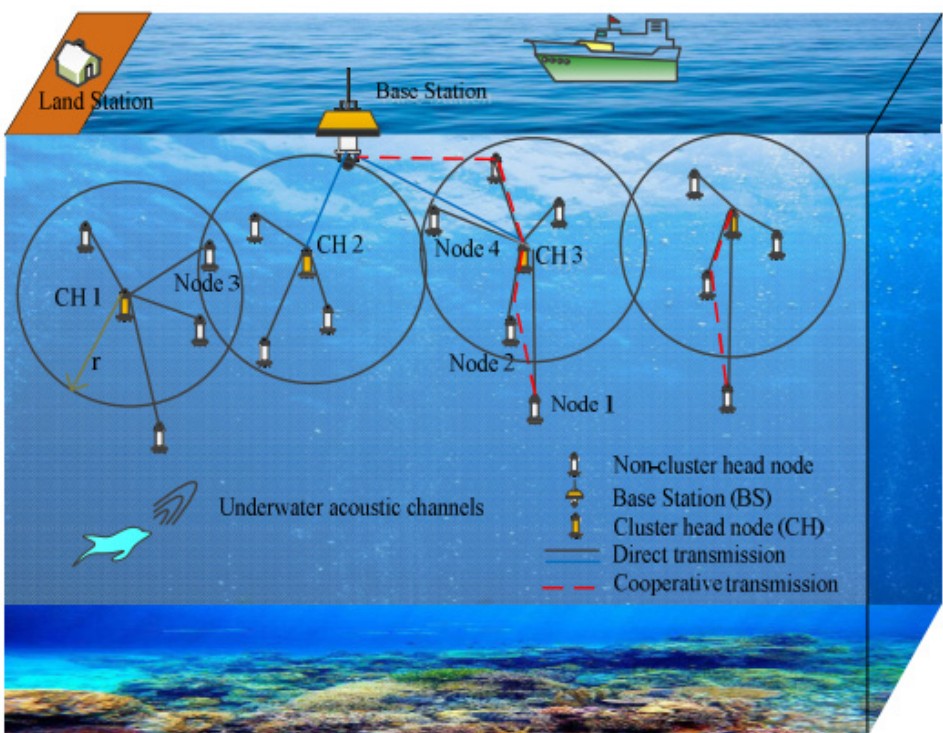

**Figure 5.** UWSN interaction model used in the MNS-CBRP approach.

To solve the issues of energy gaps, drifting underwater nodes caused by ocean currents, and insufficient bandwidth in UWSNs, we proposed a Cluster-Based Routing Protocol backed by Member Nodes (MNS-CBRP). This protocol's formal name is MNS-CBRP. The proposed MNS-CBRP technique extends the K-means method to allow for dynamic CH selection based on node distance from the centroid and residual energy [29]. The MNS-CBRP technique is used to cluster data. The Q-value function considers not only how much energy is expended by the node to transmit a packet to the CH but also how much energy is expended by the CH to send the packet to the BS. DCC is employed during the data transfer portion of the operation [30–35]. The proposed MNS-CBRP technique will also contain an ocean current model to simulate the influence of node drifting. CHs and cooperative nodes can be selected on the fly in order to enable the network to accommodate mobile nodes. The simulation results suggest that the proposed MNS-CBRP scheme has the ability to minimize total energy consumption while increasing network lifetime when compared to the current scheme [36]. Figure 5 represents a UWSN interaction model used in the MNS-CBRP approach.

In this paper, we suggest a distinct routing protocol with the purpose of concurrently improving the protocols' reliability and efficiency. This is accomplished by improving data transfer metrics, attaining optimal energy usage, and extending network lifetime. The network's performance indicators can be improved with the help of MNS-CBRP, the first of the proposed solutions [37]. A three-dimensional network was sliced into identical circles to generate the clusters. Following that, the data packets are gathered by a cluster head node, also known as an anchor node, which is responsible for gathering them from a group of randomly scattered source nodes around the network. The data packets transmitted by CHs are collected using a network comprised of a large number of surface sinks [38].

UWSNs frequently demand a large amount of power to function properly. Lowering the system's power consumption, on the other hand, not only improves the lifespan of



the sensor devices but also allows applications to run on battery power [39–42]. Inventors of battery-powered devices have more leeway for experimenting, allowing them to venture into domains where they might not otherwise succeed. Low-power wireless sensor networks have the potential to be extremely useful in these settings. Wireless sensor networks that utilize "low power" contribute to UWSN reliability by lowering the amount of power consumed by individual sensor nodes [43]. In low-power wireless sensor networks, the amount of current that may be pulled from devices while they are not in use is limited [44–47]. These constraints are intended to assist lower the total amount of electricity required to run the networks. To do this, the power statuses of the network-connected devices are altered to various states. In addition to the energy, we calculated the usual latency during transmission as well as the total amount of time spent in transmit mode [48–50]. The following is the Algorithm 1 for Cluster head Node selection.

---

**Algorithm 1:** Selection Cluster head Node (ChN) for UWSN

---

if node = ChN then
if rand $\leq$ qi then
sensor node = ChN
else
sensor node = normal node
end
else
if rand $\leq$ ri then
sensor node = ChN
else
sensor node = normal node
end
end
Where qi = smallest value of qth node
        ri = largest value of rth node

---

## 4. Simulation Results and Discussions

As part of the MNS-CBRP routing approach, an effective route is devised to permit dependable data transfer measurements at the surface sinks. Because of this limitation, the MNS-CBRP protocol requires that each source node within a circle send data packets exclusively to the cluster leader in charge of that cube. The information gathered is then transferred to the surface sinks.

After studying the performance characteristics of the 240-node MNS-CBRP network utilizing the Telnet, S-frame, and Gen-FTP applications for the purpose of this study, we came to the conclusions that are shown below in Tables 1–3. The following is a breakdown of the performance statistics for the Telnet, S-frame, and Gen-FTP programs running on the UWSN network:

**Table 1.** Parameters analysis when deploying Telnet, S-frame, and Gen-FTP of the STAR-LORA routing protocol.

| Parameter | STAR-LORA | | |
|---|---|---|---|
| | **Telnet** | **S-Frame** | **Gen-FTP** |
| Avg. txion delay (micro sec) | 58 | 60 | 64 |
| Rx power consption(mWh) | 0.15 | 0.2 | 0.019 |
| Tx power consption (mWh) | 0.16 | 0.028 | 0.14 |
| Idle power consption (mWh) | 0.85 | 0.65 | 0.68 |
| Time spent transmitting (m s) | 28 | 50 | 62 |

**Table 2.** Parameters analysis when deploying Telnet, S-frame, and Gen-FTP of the OLSR routing protocol.

| Parameter | OLSR | | |
| --- | --- | --- | --- |
| | **Telnet** | **S-Frame** | **Gen-FTP** |
| Avg. txion delay (micro sec) | 65 | 68 | 69 |
| Rx power consption(mWh) | 0.35 | 0.024 | 0.35 |
| Tx power consption (mWh) | 0.04 | 0.07 | 0.04 |
| Idle power consption (mWh) | 0.65 | 0.75 | 0.77 |
| Time spent transmitting (m s) | 16 | 17 | 22 |

**Table 3.** Parameters analysis when deploying Telnet, S-frame, and Gen-FTP of the LAR1 routing protocol.

| Parameter | LAR1 | | |
| --- | --- | --- | --- |
| | **Telnet** | **S-Frame** | **Gen-FTP** |
| Avg. txion delay (micro s) | 78 | 64 | 80 |
| Rx power consption(mWh) | 0.14 | 0.036 | 0.078 |
| Tx power consption (mWh) | 0.03 | 0.06 | 0.03 |
| Idle power consption (mWh) | 0.85 | 0.95 | 0.85 |
| Time spent transmitting (m s) | 11 | 20 | 18 |

*4.1. Available Energy When Deploying Telnet, S-Frame, and Gen-FTP in Transmit Mode of the STAR-LORA, OLSR, and LAR1 Routing Protocols*

Figure 6 is a representation of the amount of energy that is consumed by 240 nodes when STAR-LORA, OLSR, and LAR1 are used in conjunction with Telnet, S-frame, and Gen-FTP. In the case of the S-frame deployment application, the least amount of transmitting energy that is required for LAR1 is 0.006 mWh, as can be seen in Table 4.

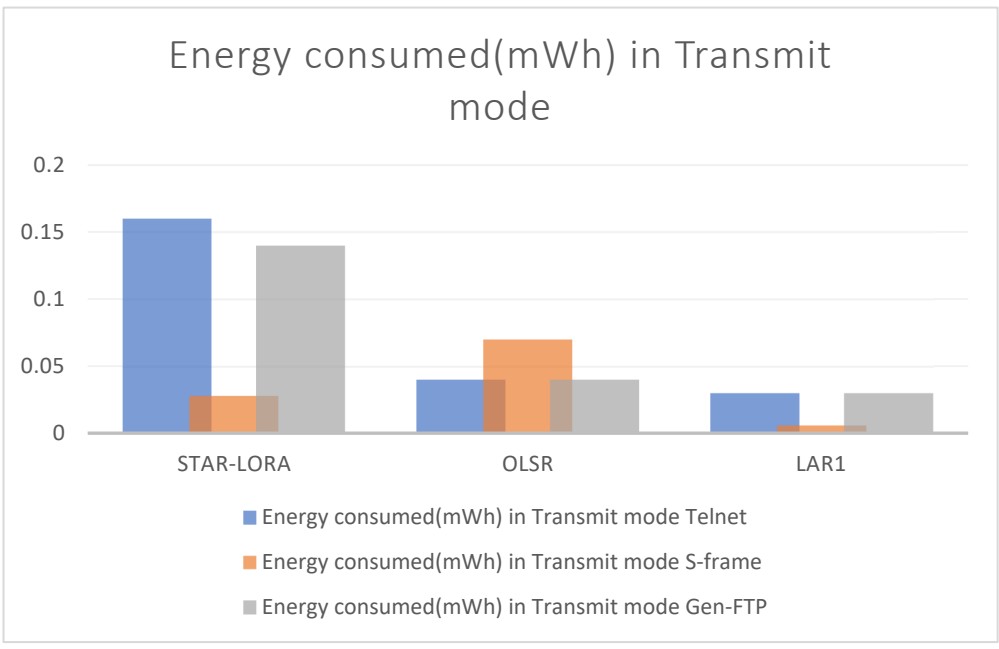

**Figure 6.** Available Energy when deploying Telnet, S-frame, and Gen-FTP in Transmit Mode of the STAR-LORA, OLSR, and LAR1 routing protocols.

**Table 4.** Parameters analysis when deploying Telnet, S-frame, and Gen-FTP of the STAR-LORA, OLSR, and LAR1 routing protocols.

| Parameter | Routing Protocol | | | | | | | | |
|---|---|---|---|---|---|---|---|---|---|
| | STAR-LORA | | | OLSR | | | LAR1 | | |
| | Telnet | S-Frame | Gen-FTP | Telnet | S-Frame | Gen-FTP | Telnet | S-Frame | Gen-FTP |
| Avg. txion delay (micro s) | 58 | 60 | 64 | 65 | 68 | 69 | 78 | 64 | 80 |
| Rx power consption(mWh) | 0.15 | 0.2 | 0.019 | 0.35 | 0.024 | 0.35 | 0.14 | 0.036 | 0.078 |
| Tx power consption (mWh) | 0.16 | 0.028 | 0.14 | 0.04 | 0.07 | 0.04 | 0.03 | 0.006 | 0.03 |
| Idle power consption (mWh) | 0.85 | 0.65 | 0.68 | 0.65 | 0.75 | 0.77 | 0.85 | 0.95 | 0.85 |
| Time spent transmitting (m s) | 28 | 50 | 62 | 16 | 17 | 22 | 11 | 20 | 18 |

*4.2. Available Energy When Deploying Telnet, S-Frame, and Gen-FTP in Receive Mode of the STAR-LORA, OLSR, and LAR1 Routing Protocols*

Figure 7 is a representation of the amount of energy that is consumed by 240 nodes when STAR-LORA, OLSR, and LAR1 are used in conjunction with Telnet, S-frame, and Gen-FTP. Table 4 illustrates that the minimum amount of receiving energy required for STAR-LORA in the Gen-FTP deployment application is 0.019 mWh. This information may be found in Table 4.

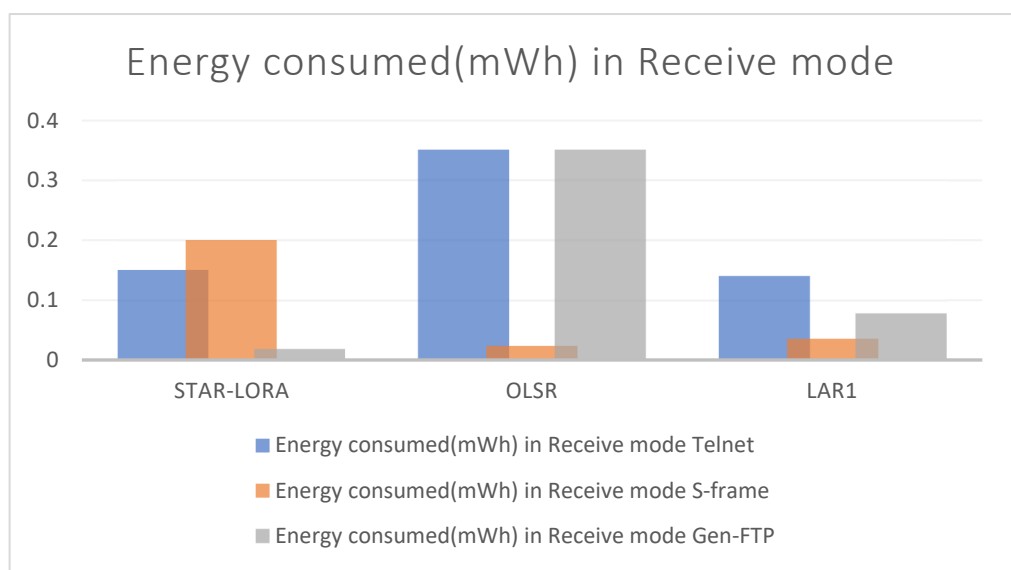

**Figure 7.** Available Energy when deploying Telnet, S-frame, and Gen-FTP in Receive Mode of the STAR-LORA, OLSR, and LAR1 routing protocols.

*4.3. Available Energy When Deploying Telnet, S-Frame, and Gen-FTP in Idle Mode of the STAR-LORA, OLSR, and LAR1 Routing Protocols*

Figure 8 illustrates the amount of power that is consumed by 240 nodes when STAR-LORA, OLSR, and LAR1 are used in conjunction with Telnet, S-frame, and Gen-FTP. Table 4 illustrates that the minimum amount of idle energy necessary for STAR-LORA in the S-frame deployment application is 0.65 mWh and that the minimum amount of idle energy required for OLSR in the Telnet deployment application is 0.65 mWh.

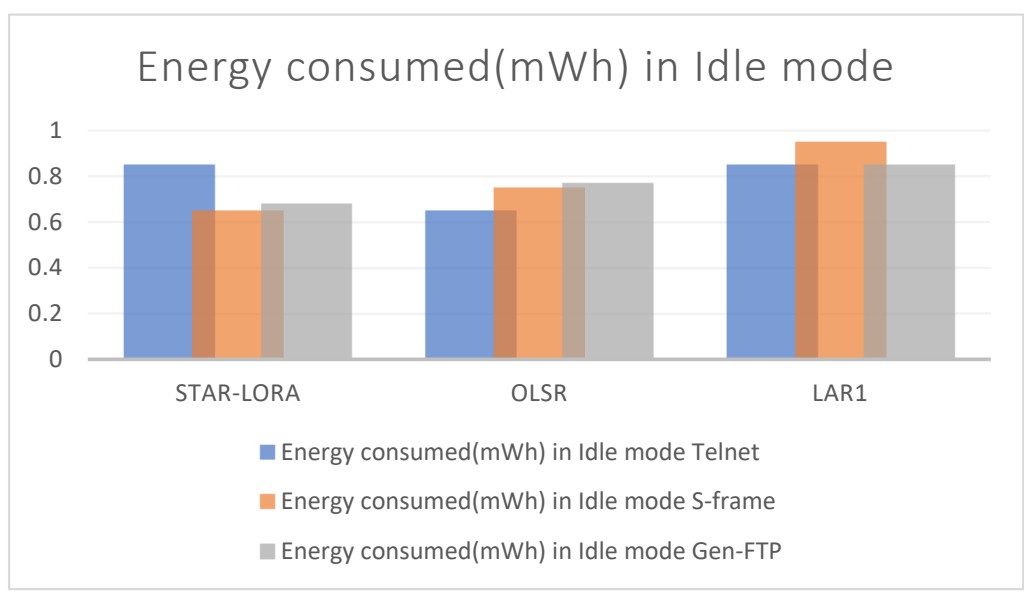

**Figure 8.** Available Energy when deploying Telnet, S-frame, and Gen-FTP in Idle Mode of the STAR-LORA, OLSR, and LAR1 routing protocols.

*4.4. Avg. Txion Delay (Micro Sec) When Deploying Telnet, S-Frame, and Gen-FTP of the STAR-LORA, OLSR, and LAR1 Routing Protocols*

Figure 9 illustrates the typical transmission delay (in microseconds) experienced by 240 nodes when STAR-LORA, OLSR, and LAR1 are used in conjunction with Telnet, S-frame, and Gen-FTP. Table 4 illustrates that the smallest amount of minimum necessary average transmission time for STAR-LORA in the Telnet deployment application is 58 microseconds.

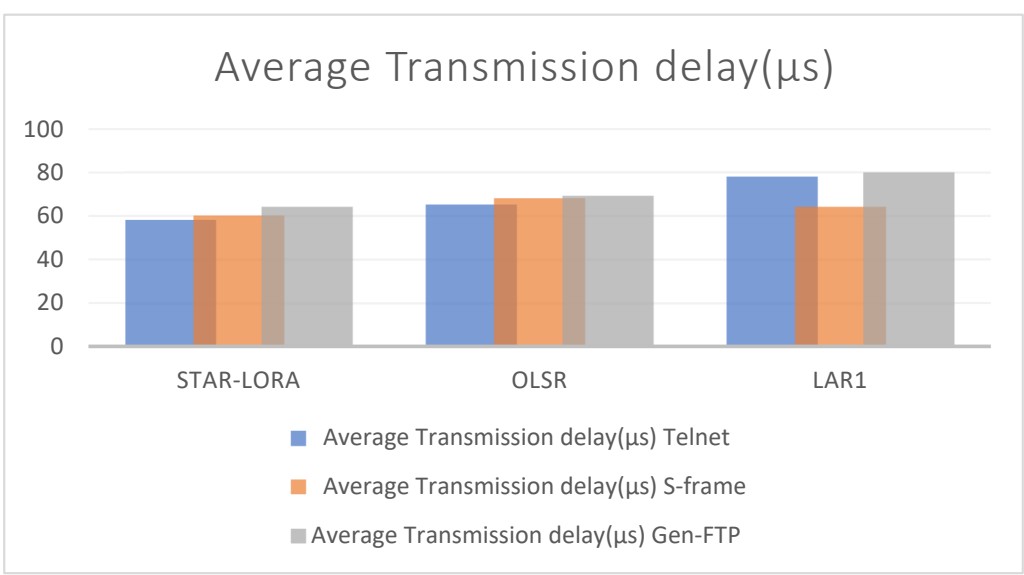

**Figure 9.** Avg. Txion delay (micro sec) when deploying Telnet, S-frame, and Gen-FTP of the STAR-LORA, OLSR, and LAR1 routing protocols.

*4.5. Time Spent Transmitting (m s) When Deploying Telnet, S-Frame, and Gen-FTP of the STAR-LORA, OLSR, and LAR1 Routing Protocols*

Figure 10 illustrates the amount of power that is consumed by 240 nodes when STAR-LORA, OLSR, and LAR1 are used in conjunction with Telnet, S-frame, and Gen-FTP. Table 4 demonstrates that 11 milliseconds is the minimum amount of time that must be spent sending in order to meet the requirements for LAR1 in the Telnet deployment application.

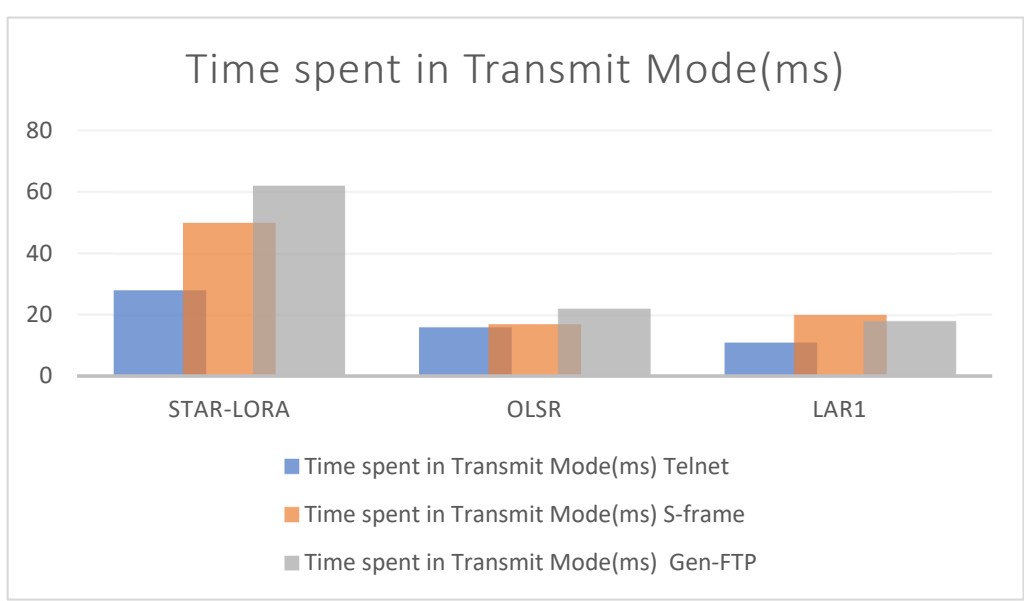

**Figure 10.** Time spent transmitting (m s) when deploying Telnet, S-frame, and Gen-FTP of the STAR-LORA, OLSR, and LAR1 routing protocols.

Figures A1–A15 illustrate the conclusions of the study, which include the utilization of available energy during the deployment of Telnet. Figures A16–A30 illustrate the conclusions of the study, which include the utilization of available energy during the deployment of the S-frame. Figures A31–A45 illustrate the conclusions of the study, which include the utilization of available energy during the deployment of Gen-FTP for a variety of routing protocols, such as STAR-LORA, OLSR, and LAR1.

## 5. Conclusions

The study of the ocean floor is linked with data analysis, marine life monitoring, and military planning. This is because being underwater is required for all three of these activities. Because of the limits imposed on the network by its restricted capabilities, the UWSN's battery life is prioritized. Several common routing protocols, such as telnet, S-frame, and Gen-FTP, are analyzed and compared in UWSN networks with varying deployment conditions. One of the several measures investigated was energy consumption during transmission, standby, and reception and also the average transmission delay time spent for transmitting bytes. The proposed MNS-CBRP employs the LAR1 routing protocol, which spends 0.006 mWh of energy in the S-frame deployment application's transmit mode, and the STAR-LORA routing protocol, which consumes 0.019 mWh of energy in the Gen-FTP deployment application's receive mode and the OLSR routing protocol, which consumes 0.65 mWh of energy in the Telnet deployment application's receive mode. Furthermore, the transmission latency for UWSN using STAR-LORA is sixty microseconds for the Telnet deployment application. Additionally, for the Telnet deployment application, LAR1 spends 11 msec on the Telnet deployment application.

**Author Contributions:** Conceptualization, methodology, writing—original draft, results analysis, K.S.; data collection, data analysis, writing—review and editing, results analysis, M.H.; methodology, writing—review and editing, design and presentation, references, R.C.; methodology, writing—review and editing, G.P.; methodology, writing—review and editing, A.K.; methodology, writing—review and editing, R.A.; methodology, writing—review and editing, M.A.; methodology, writing—review and editing, M.U. All authors have read and agreed to the published version of the manuscript.

**Funding:** This research was funded by Princess Nourah bint Abdulrahman University Researchers Supporting Project number (PNURSP2023R125), Princess Nourah bint Abdulrahman University, Riyadh, Saudi Arabia. This work was also supported by King Khalid University (KKU) through the Research Group Program Under the Grant Number: (R.G.P.1/224/43).

**Institutional Review Board Statement:** Not applicable.

**Informed Consent Statement:** Not applicable.

**Data Availability Statement:** The datasets used during the current study are available from the corresponding author upon reasonable request.

**Acknowledgments:** This research was funded by Princess Nourah bint Abdulrahman University Researchers Supporting Project number (PNURSP2023R125), Princess Nourah bint Abdulrahman University, Riyadh, Saudi Arabia. The authors extend their appreciation to the Deanship of Scientific Research at King Khalid University (KKU) for funding this work through the Research Group Program Under the Grant Number: (R.G.P.1/224/43).

**Conflicts of Interest:** The authors declare no conflict of interest.

## List of Abbreviations

| | |
|---|---|
| UWSNs | Underwater Wireless Sensor Networks |
| UAWSNs | Underwater Acoustic Wireless Sensor Networks |
| MNS-CBRP | Member Nodes Supported Cluster-Based Routing Protocol |
| CH | Cluster Head |
| IoUT | Internet of Underwater Things |
| IoT | Internet of Things |
| PLUSNet | Persistent Littoral Undersea Surveillance Network |
| AUVs | Autonomous Underwater Vehicles |
| ROVs | Remotely Operated Vehicles. |
| TWSN | Terrestrial Wireless Sensor Networks |
| LEACH | Low-Energy Adaptive Clustering Hierarchy |
| GAF | Group Adaptive Filtering |
| HEED | Hybrid Energy Efficient Distributed Clustering |
| GPS | Global Positioning System |
| BGAF | Based on the GAF algorithm |
| STAR-LORA | Source Tree Adaptive Routing-Least Overhead Routing Approach |
| CBR | Constant Bit Rate |
| Telnet | Teletype network Protocol |
| S-frame | Supervisory frame |
| Gen-FTP | Generic File Transfer Protocol |
| OLSR | Optimized Link State Routing |
| LAR1 | Location-Aided Routing |
| BS | Base Station |
| DCC | Dynamic Coded Collaboration |
| ChN | Cluster head Node |

## Appendix A

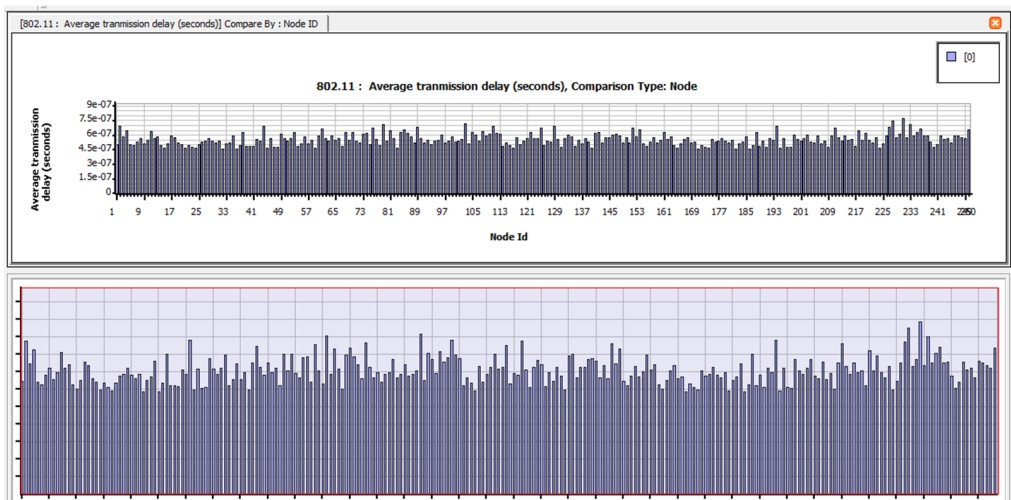

**Figure A1.** The average time delay required for transmissions during a Telnet deployment of STAR-LORA.

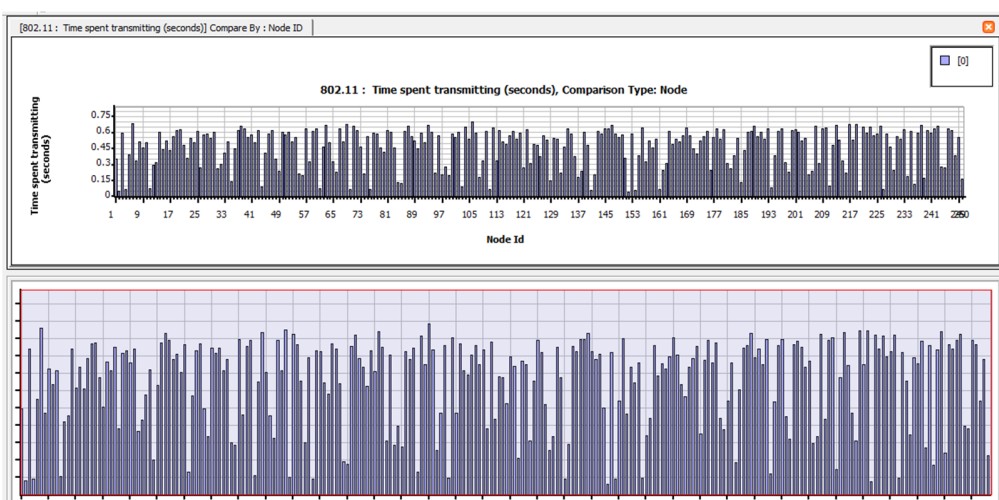

**Figure A2.** Transmission time during the deployment of Telnet to the STAR-LORA.

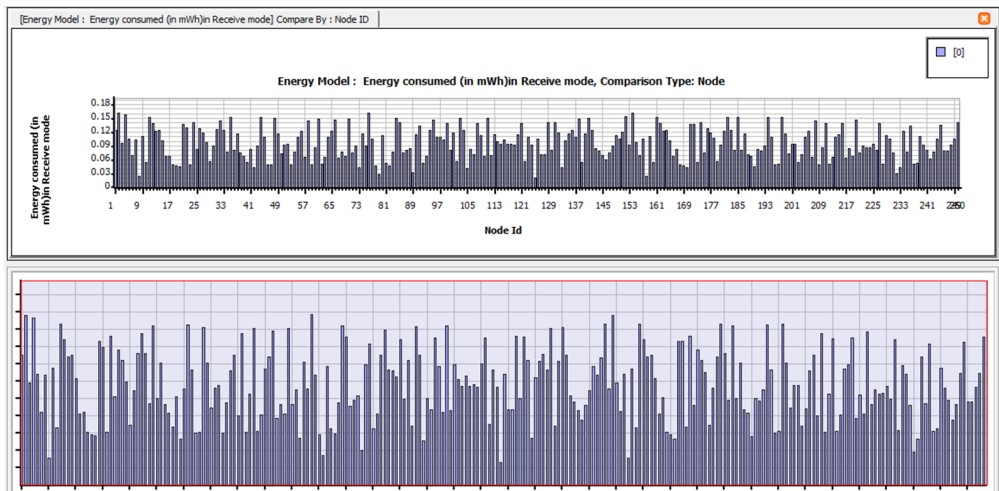

**Figure A3.** Telnet deployment can be accomplished by utilizing the energy available when STAR-LORA is in its Receive mode.

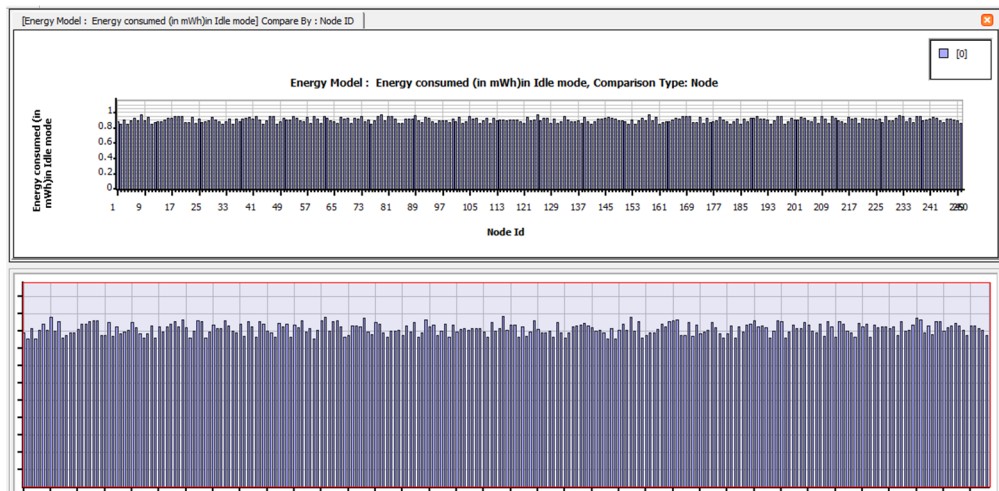

**Figure A4.** Telnet deployment can be accomplished by utilizing the energy available when STAR-LORA is in its idle mode.

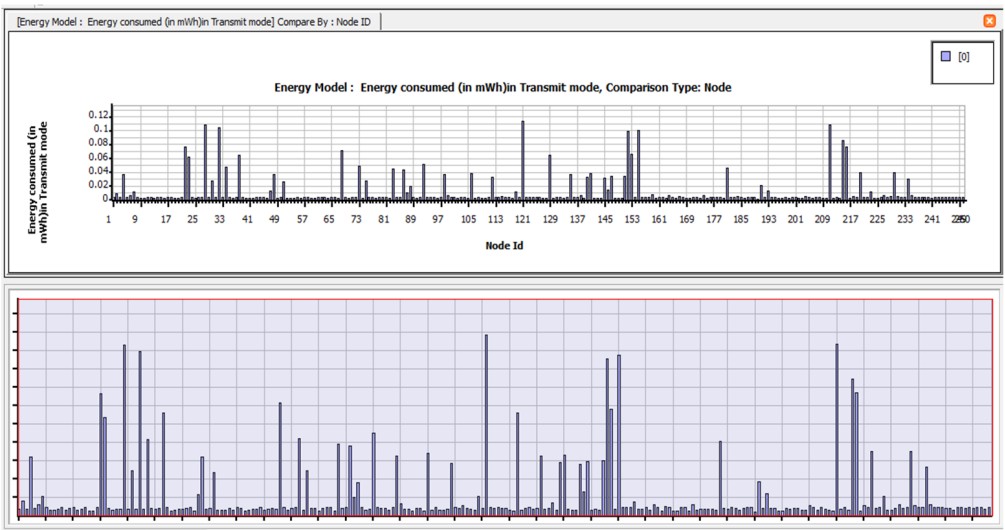

**Figure A5.** Telnet deployment can be accomplished by utilizing the energy available when STAR-LORA is in its Transmit mode.

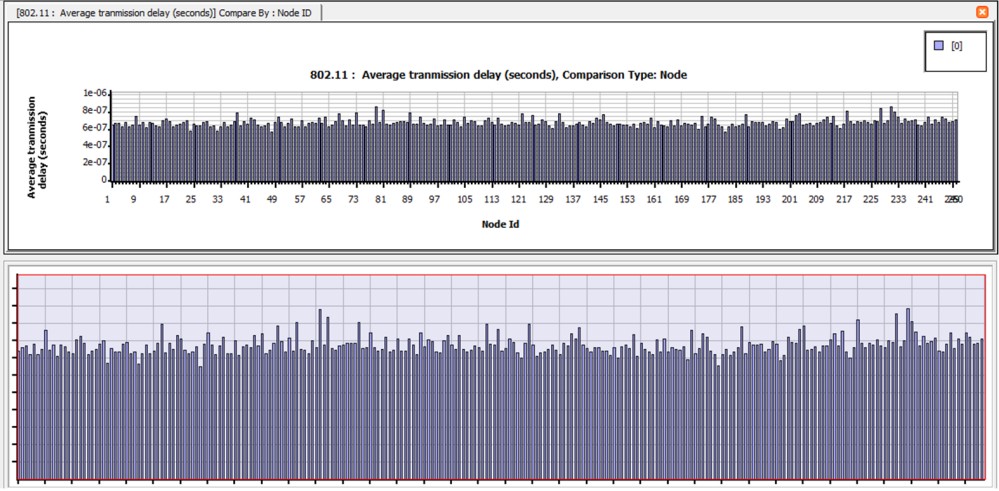

**Figure A6.** The average time delay required for transmissions during a Telnet deployment of OLSR.

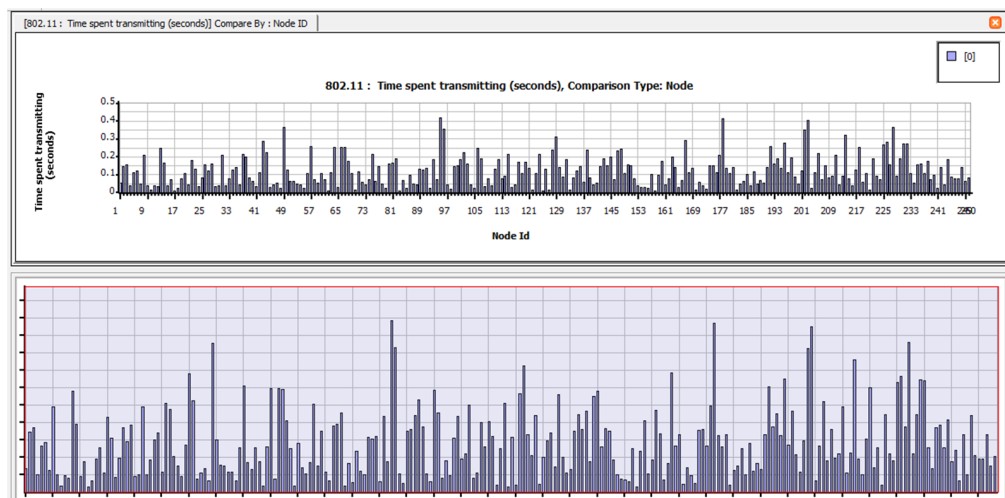

**Figure A7.** Transmission time during the deployment of Telnet to the OLSR.

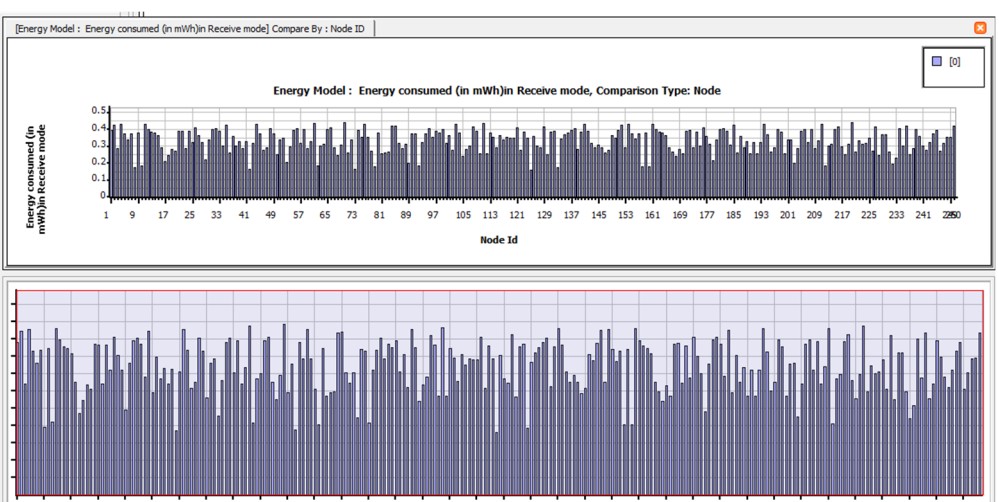

**Figure A8.** Telnet deployment can be accomplished by utilizing the energy available when OLSR is in its Receive mode.

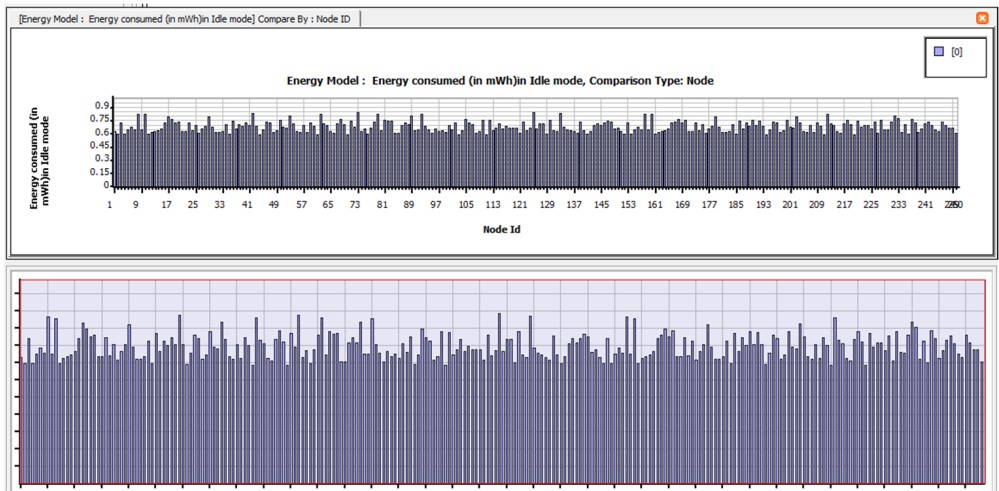

**Figure A9.** Telnet deployment can be accomplished by utilizing the energy available when OLSR is in its idle mode.

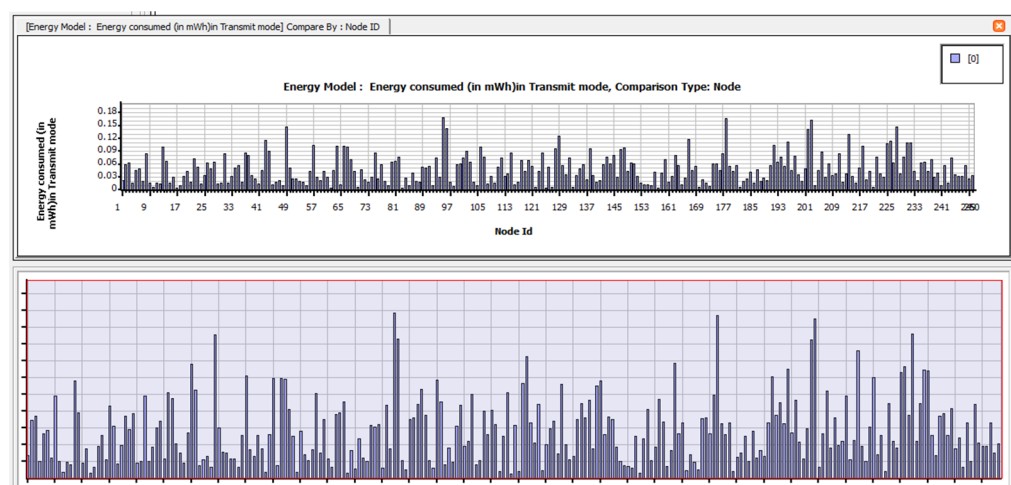

**Figure A10.** Telnet deployment can be accomplished by utilizing the energy available when OLSR is in its Transmit mode.

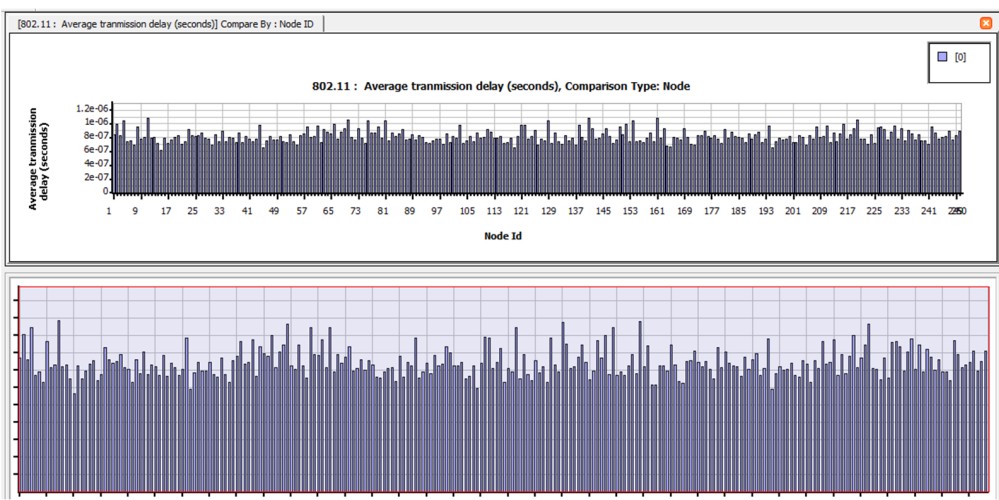

**Figure A11.** The average time delay required for transmissions during a Telnet deployment of LAR1.

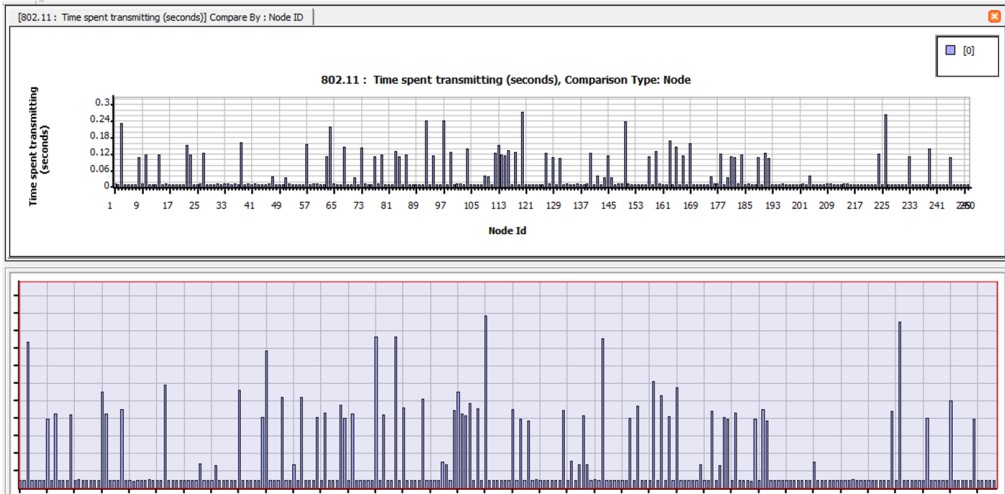

**Figure A12.** Transmission time during the deployment of Telnet to the LAR1.

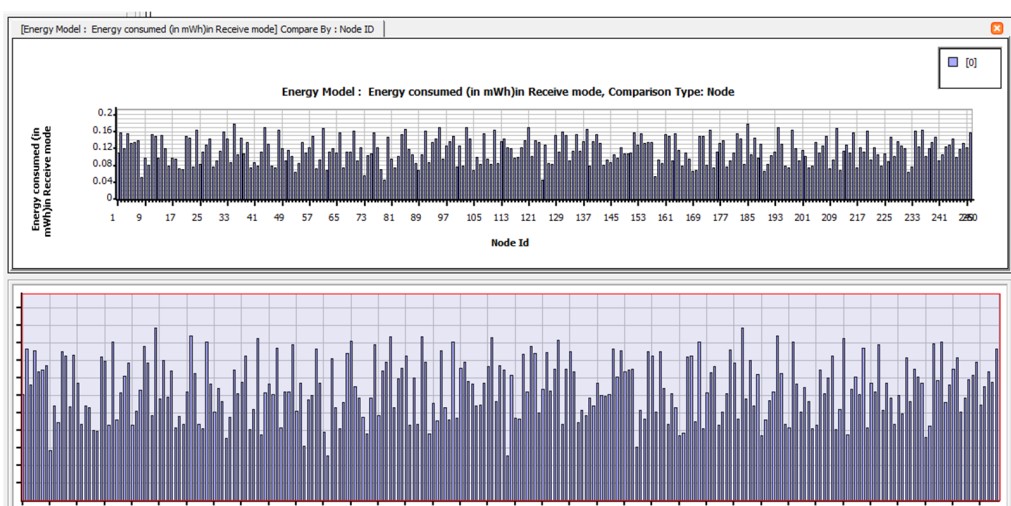

**Figure A13.** Telnet deployment can be accomplished by utilizing the energy available when LAR1 is in its Receive mode.

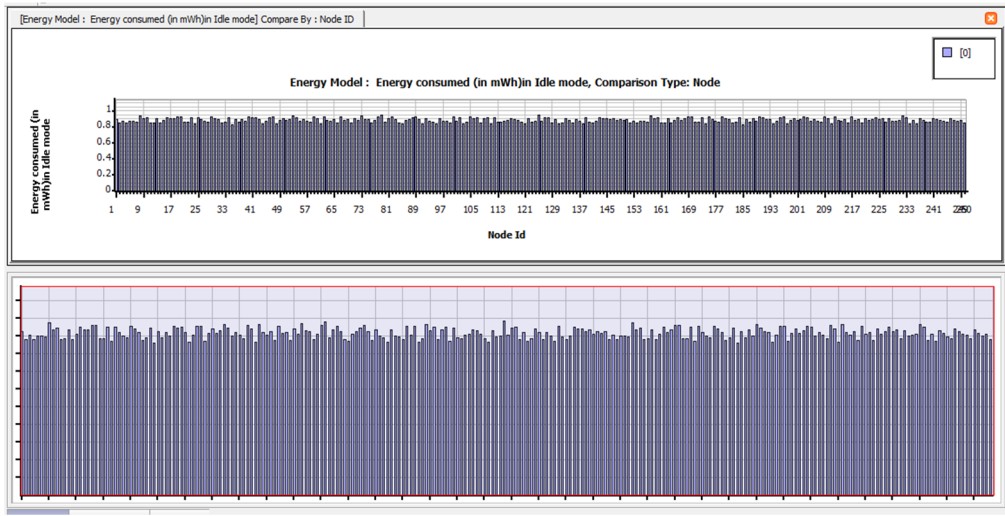

**Figure A14.** Telnet deployment can be accomplished by utilizing the energy available when LAR1 is in its idle mode.

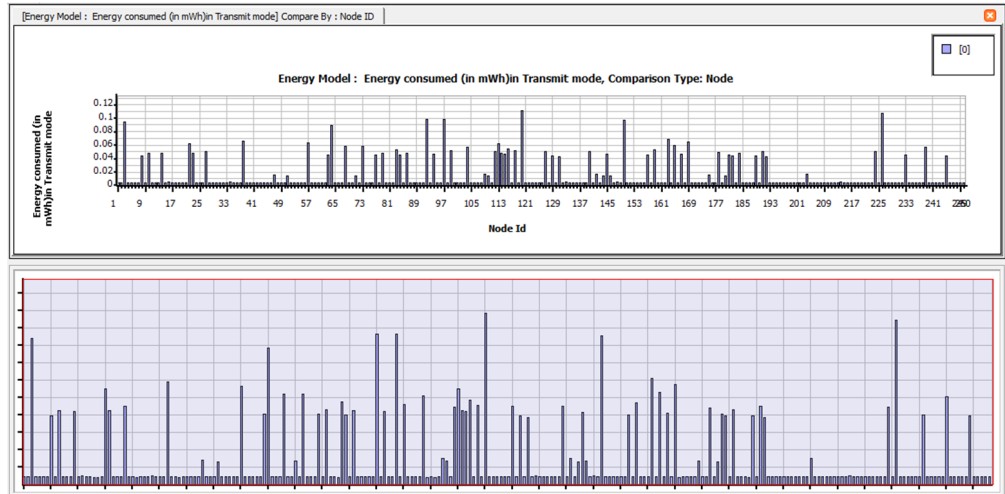

**Figure A15.** Telnet deployment can be accomplished by utilizing the energy available when LAR1 is in its Transmit mode.

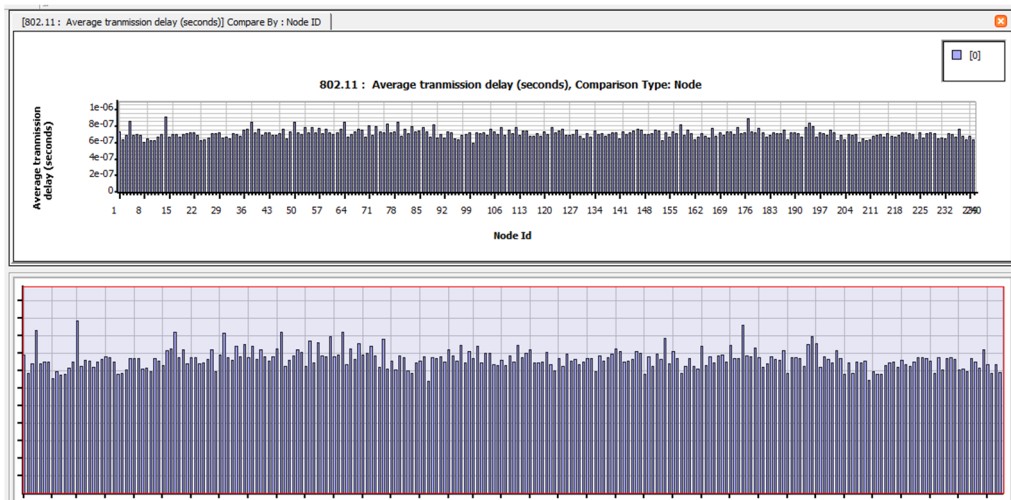

**Figure A16.** The average time delay required for transmissions during an S-frame deployment of STAR-LORA.

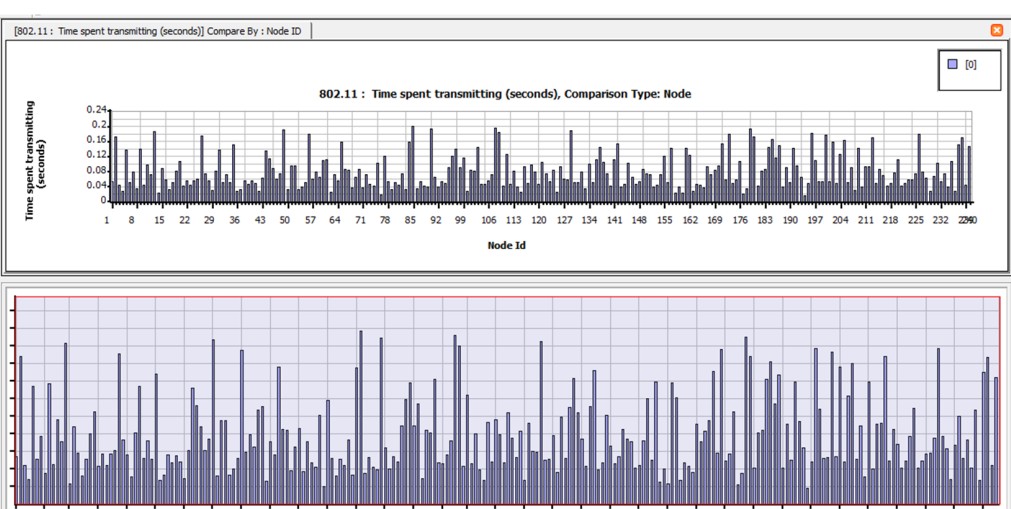

**Figure A17.** Transmission time during the deployment of the S-frame to the STAR-LORA.

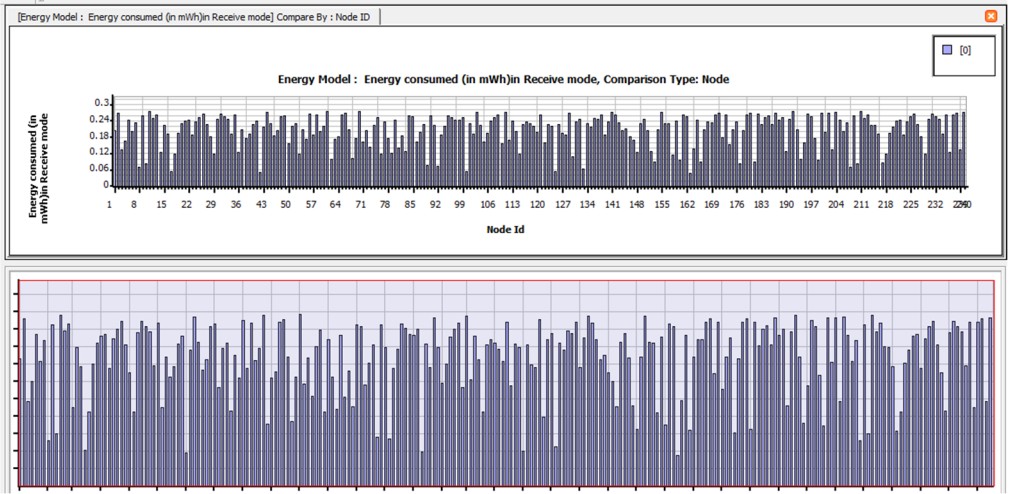

**Figure A18.** S-frame deployment can be accomplished by utilizing the energy available when STAR-LORA is in its Receive mode.

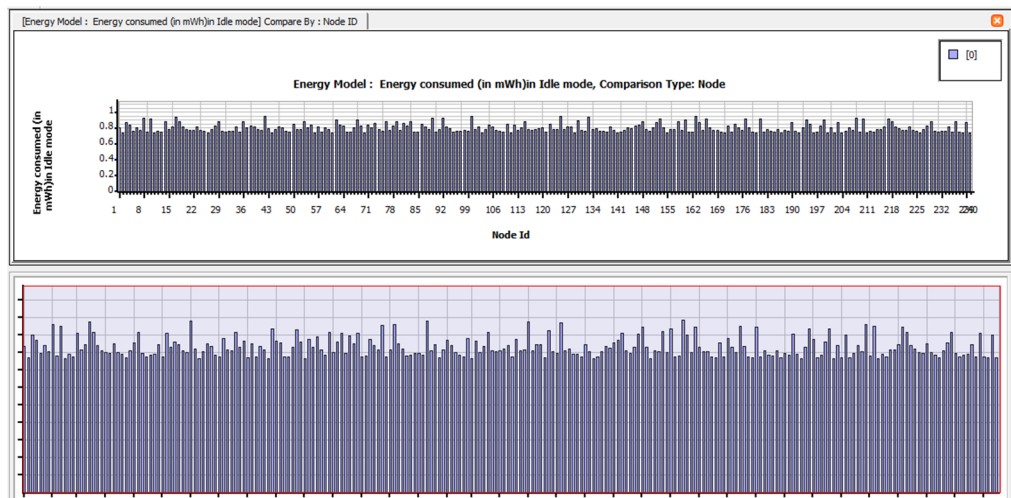

**Figure A19.** S-frame deployment can be accomplished by utilizing the energy available when STAR-LORA is in its idle mode.

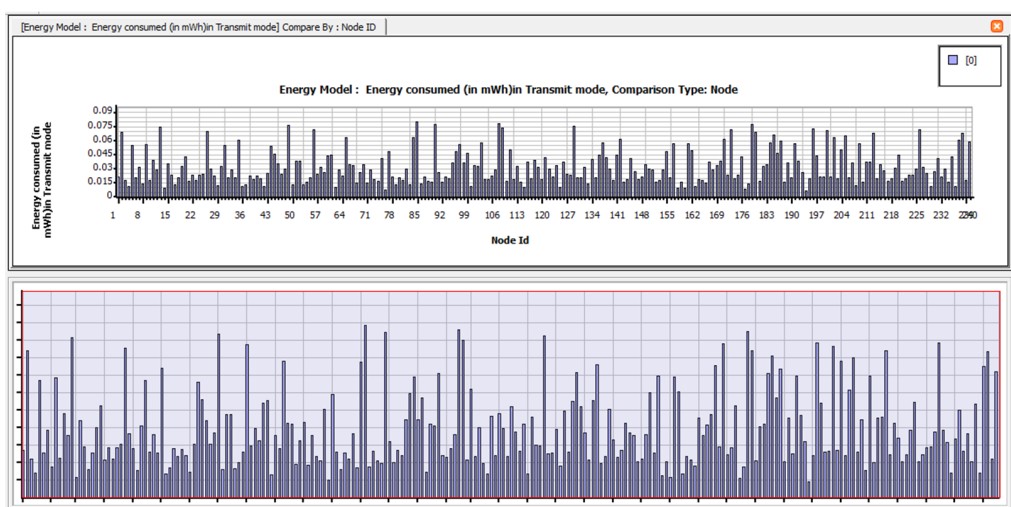

**Figure A20.** S-frame deployment can be accomplished by utilizing the energy available when STAR-LORA is in its Transmit mode.

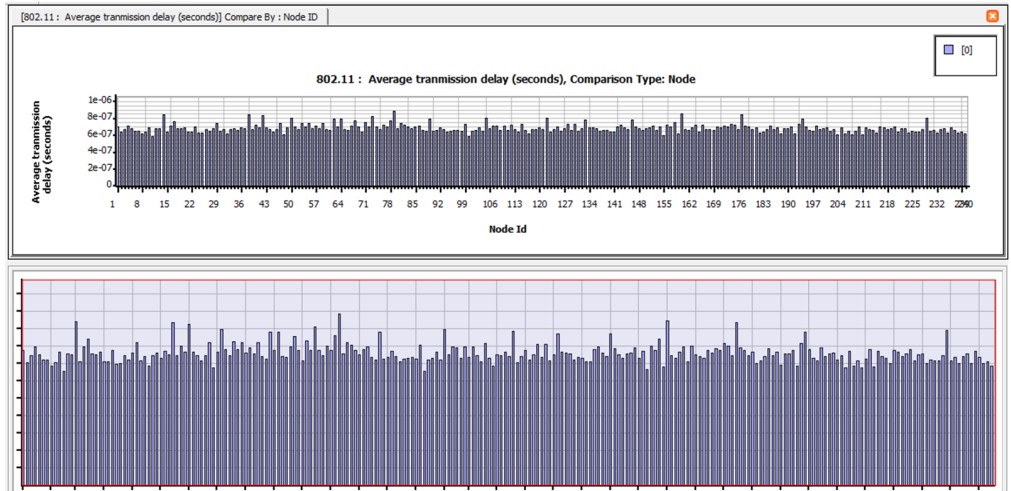

**Figure A21.** The average time delay required for transmissions during an S-frame deployment of OLSR.

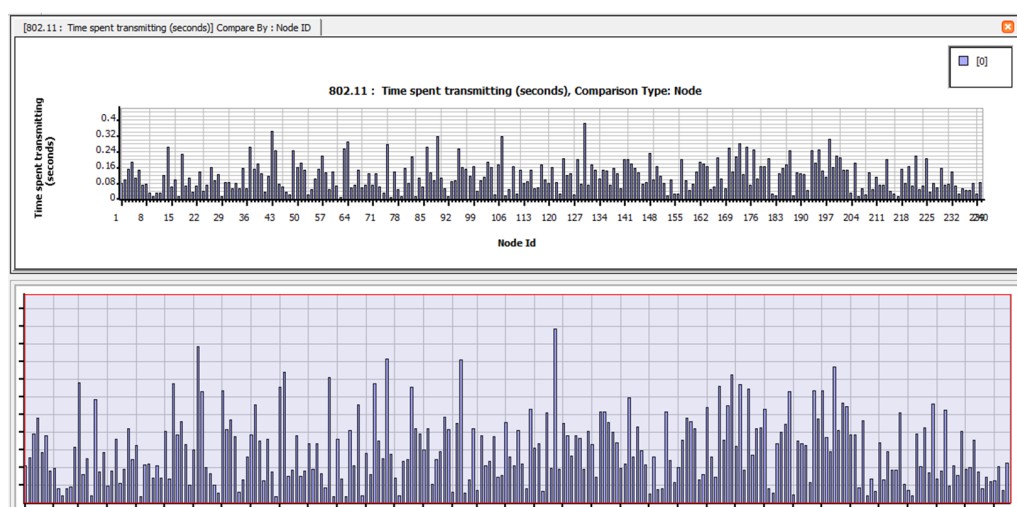

**Figure A22.** Transmission time during the deployment of the S-frame to the OLSR.

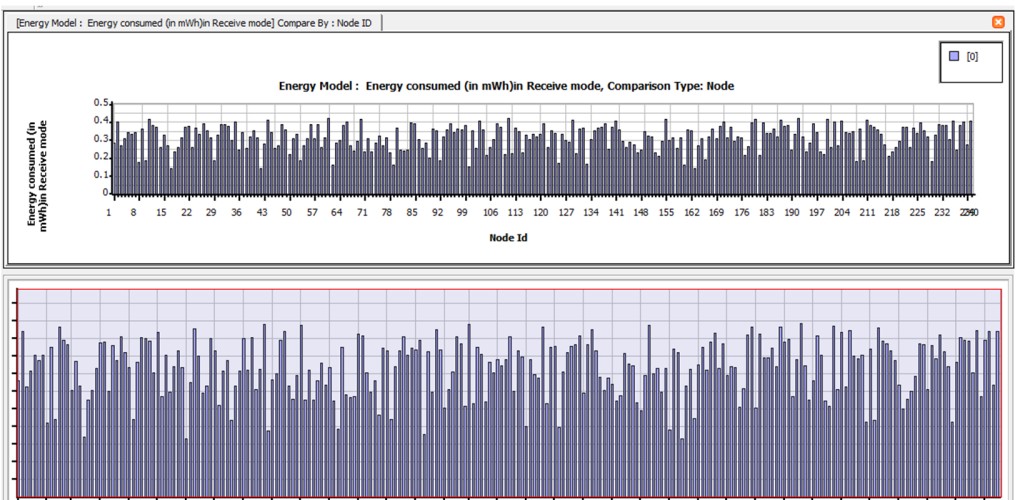

**Figure A23.** S-frame deployment can be accomplished by utilizing the energy available when OLSR is in its Receive mode.

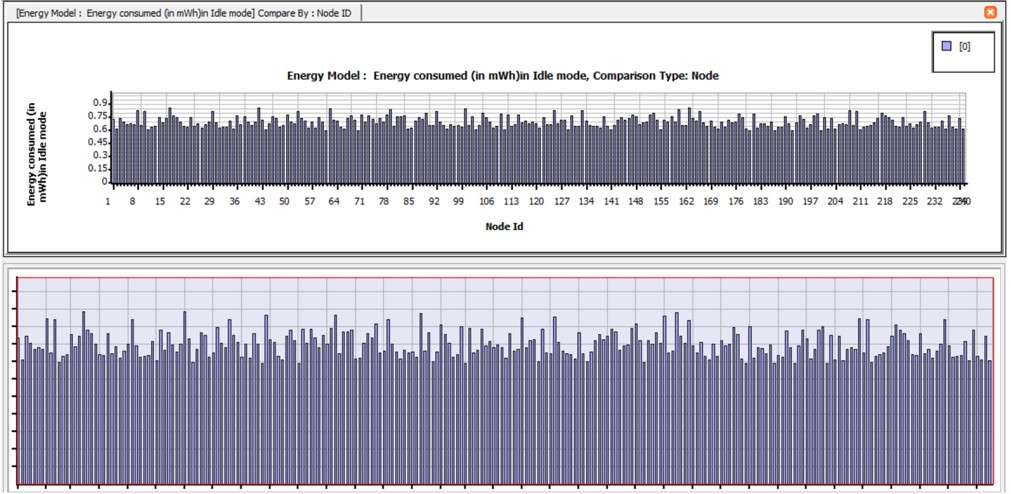

**Figure A24.** S-frame deployment can be accomplished by utilizing the energy available when OLSR is in its idle mode.

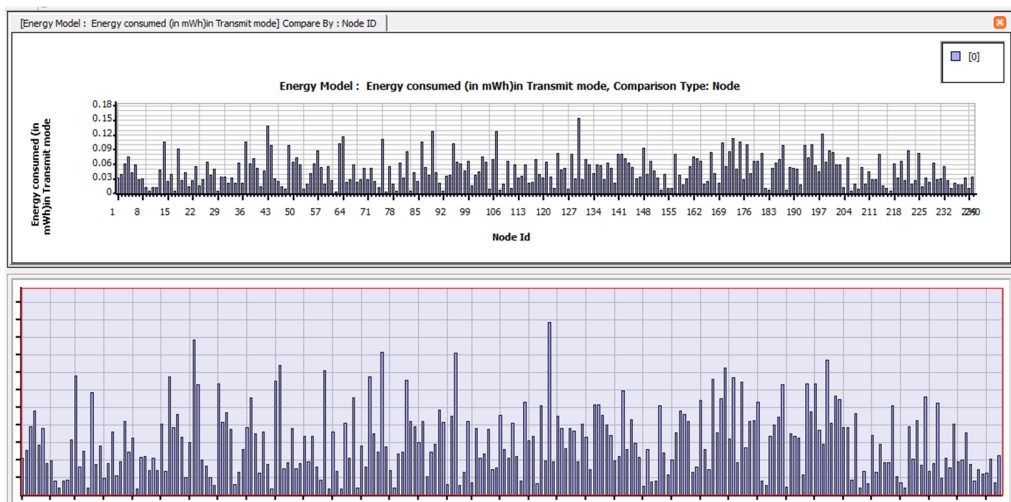

**Figure A25.** S-frame deployment can be accomplished by utilizing the energy available when OLSR is in its Transmit mode.

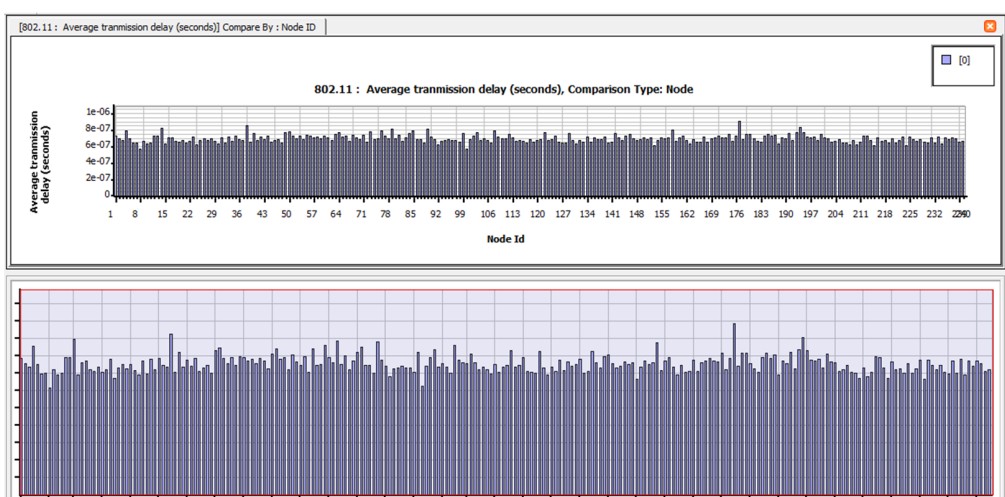

**Figure A26.** The average time delay required for transmissions during an S-frame deployment of LAR1.

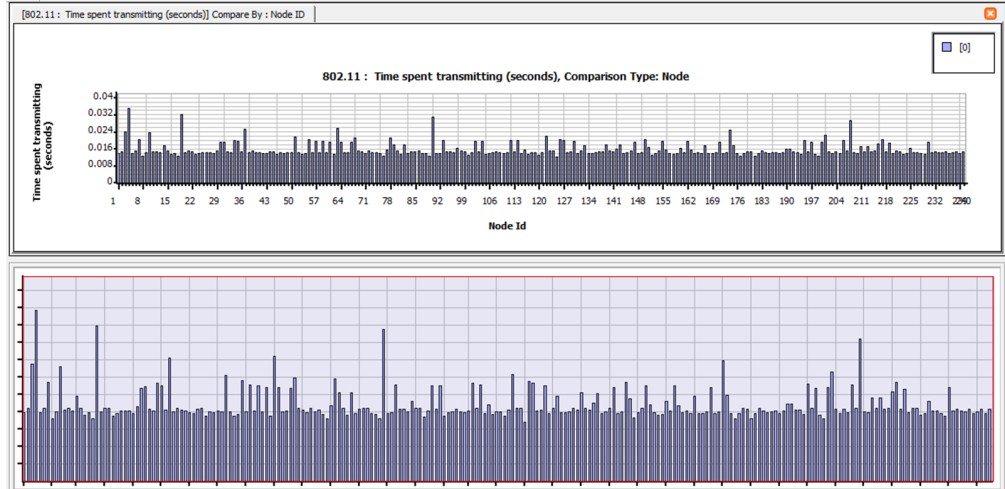

**Figure A27.** Transmission time during the deployment of the S-frame to the LAR1.

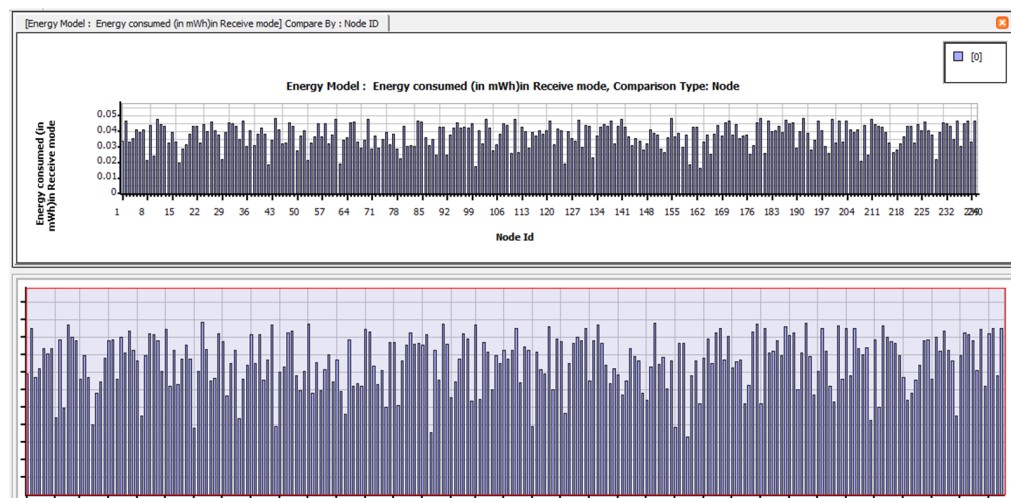

**Figure A28.** S-frame deployment can be accomplished by utilizing the energy available when LAR1 is in its Receive mode.

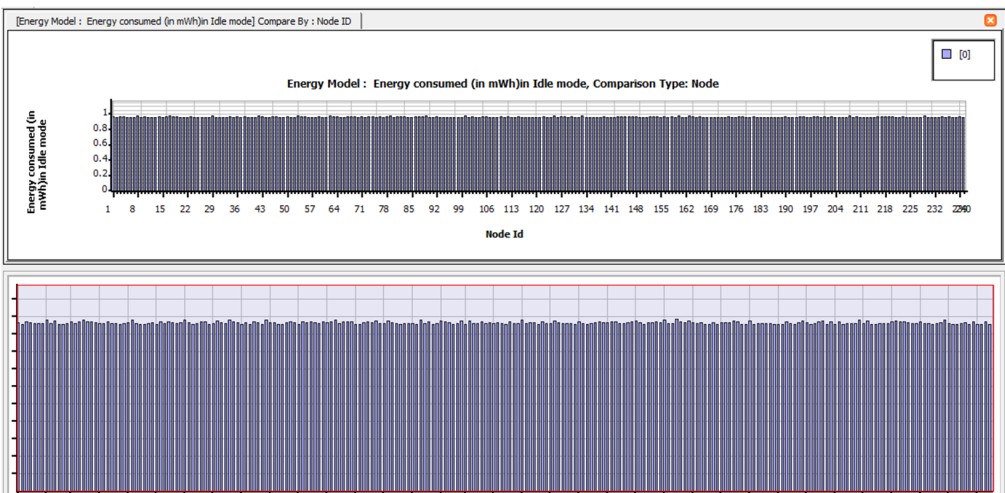

**Figure A29.** S-frame deployment can be accomplished by utilizing the energy available when LAR1 is in its idle mode.

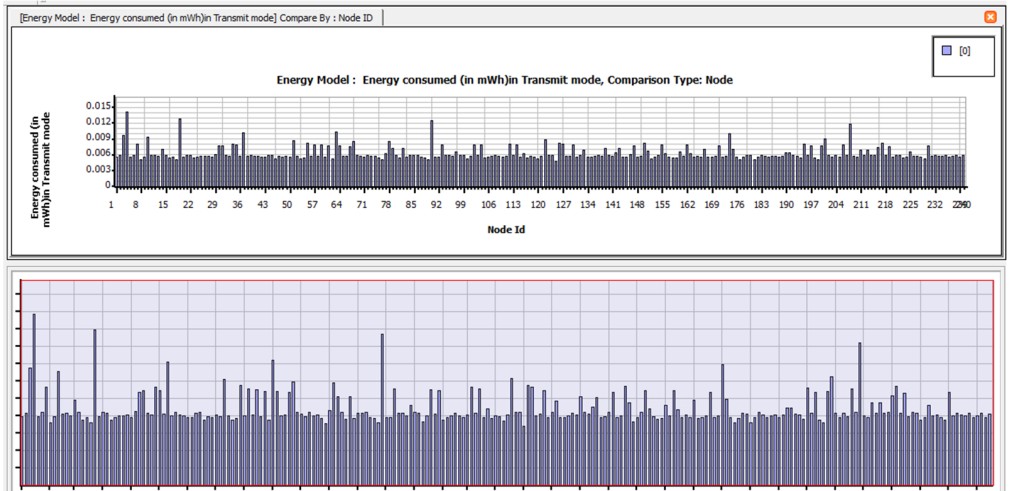

**Figure A30.** S-frame deployment can be accomplished by utilizing the energy available when LAR1 is in its Transmit mode.

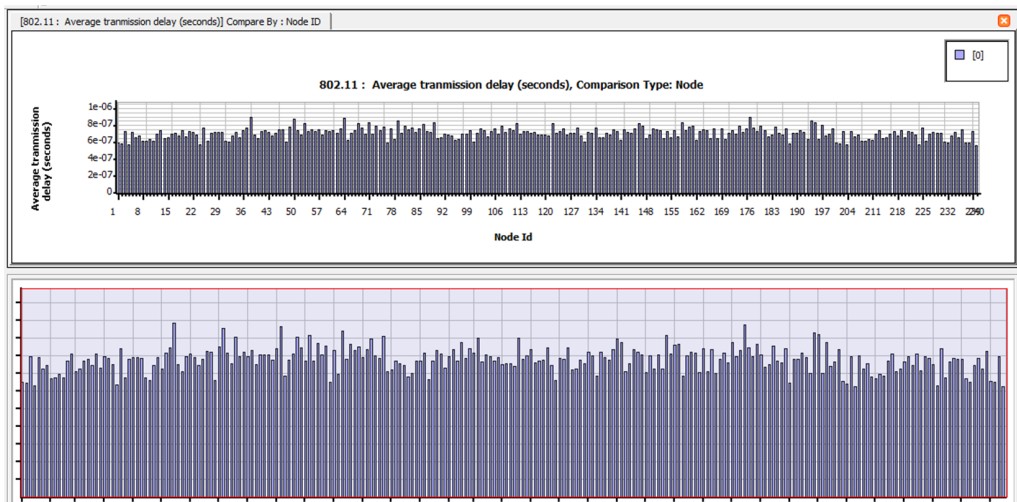

**Figure A31.** The average time delay required for transmissions during a Gen-FTP deployment of STAR-LORA.

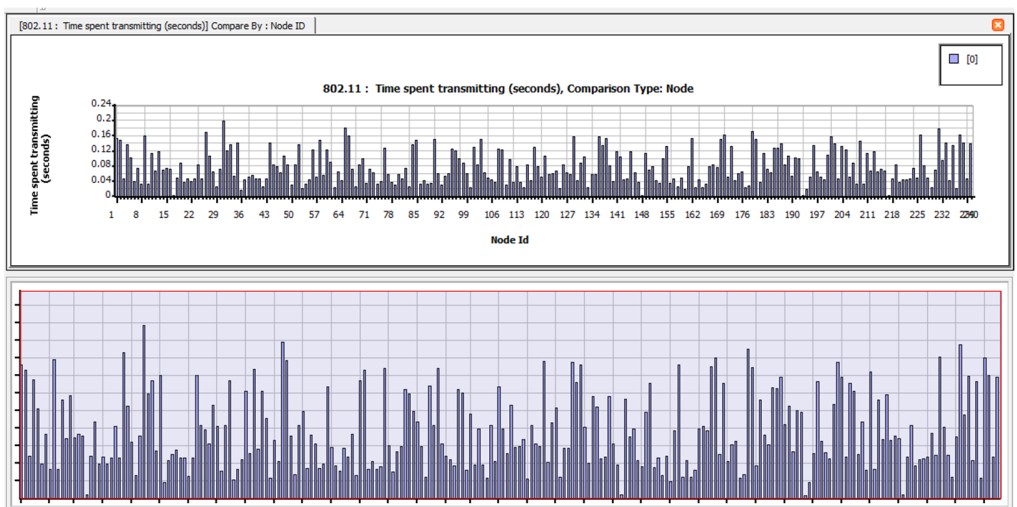

**Figure A32.** Transmission time during the deployment of the Gen-FTP to the STAR-LORA.

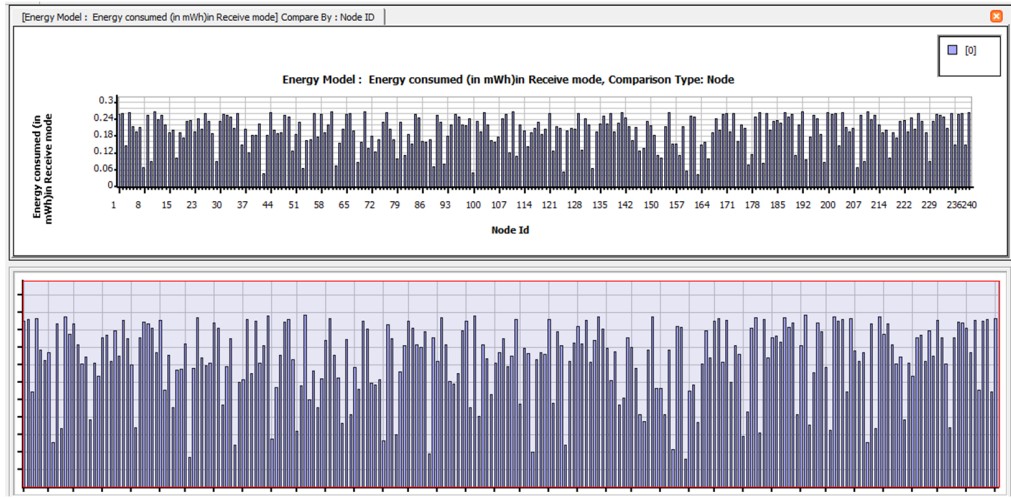

**Figure A33.** Gen-FTP deployment can be accomplished by utilizing the energy available when STAR-LORA is in its Receive mode.

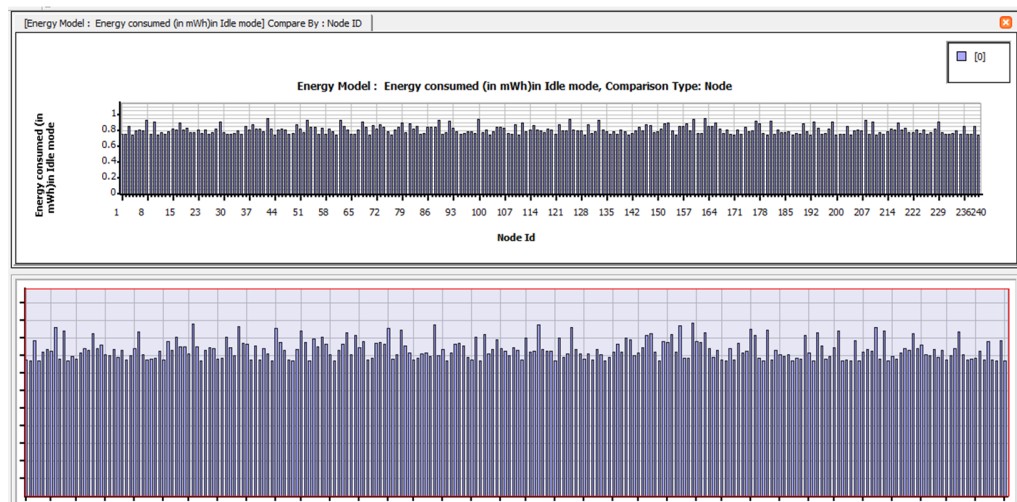

**Figure A34.** Gen-FTP deployment can be accomplished by utilizing the energy available when STAR-LORA is in its idle mode.

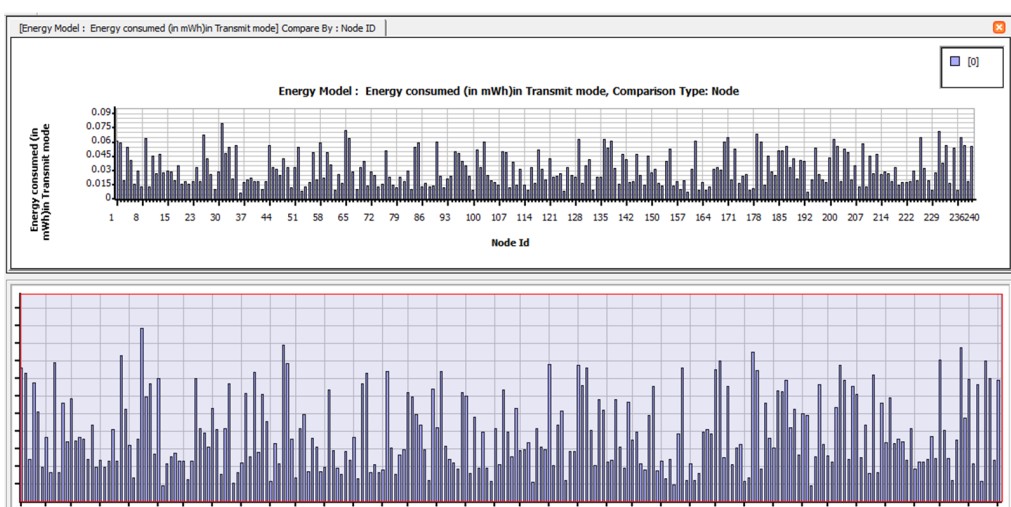

**Figure A35.** Gen-FTP deployment can be accomplished by utilizing the energy available when STAR-LORA is in its Transmit mode.

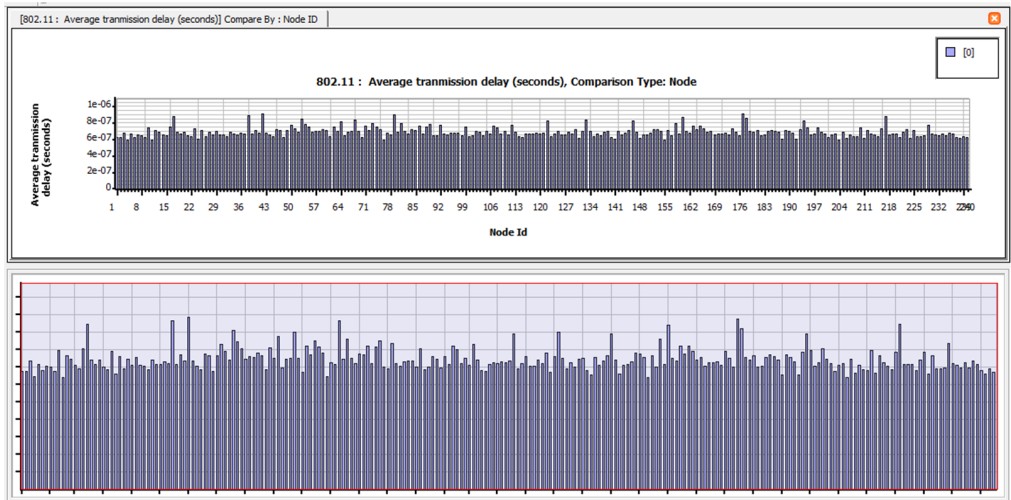

**Figure A36.** The average time delay required for transmissions during a Gen-FTP deployment of OLSR.

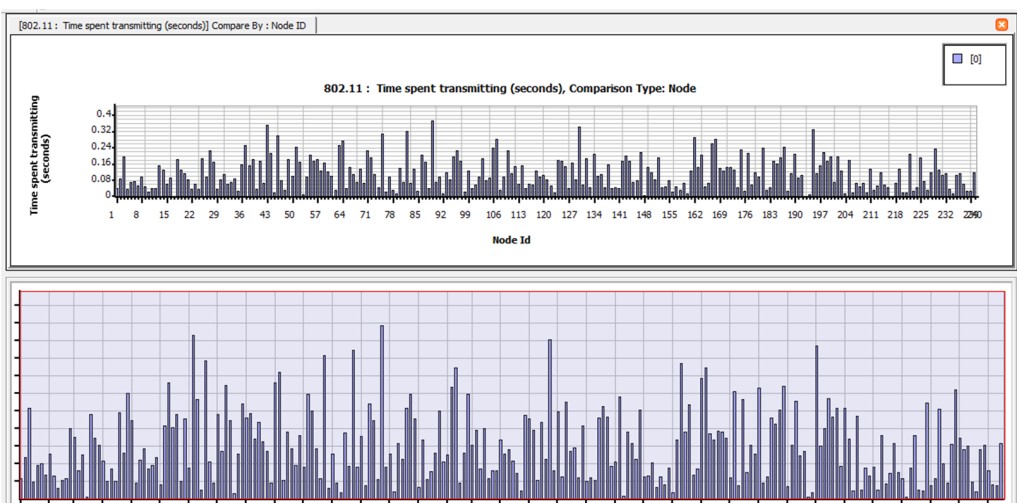

**Figure A37.** Transmission time during the deployment of the Gen-FTP to the OLSR.

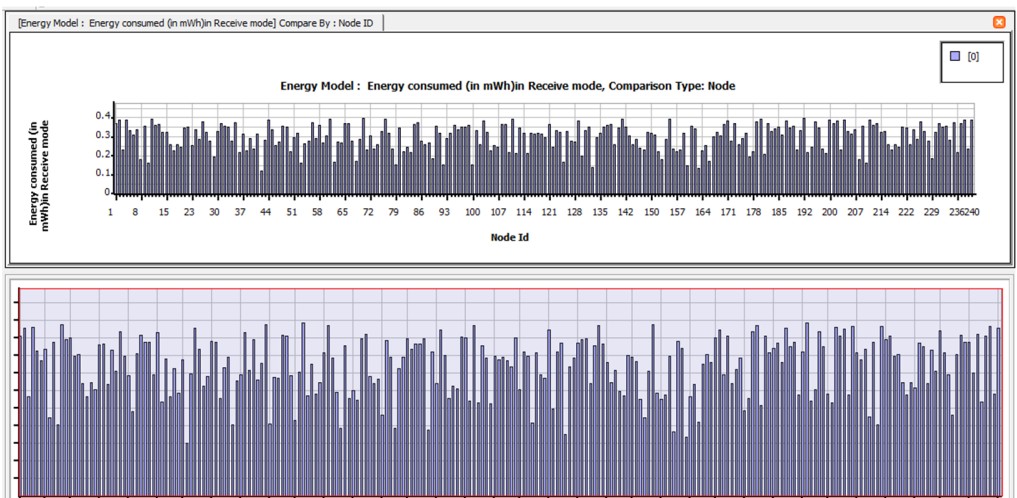

**Figure A38.** Gen-FTP deployment can be accomplished by utilizing the energy available when OLSR is in its Receive mode.

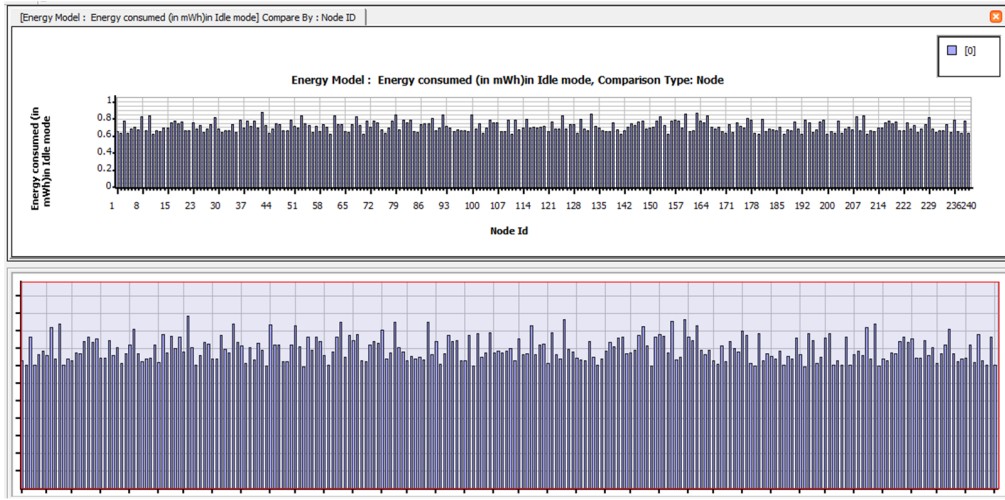

**Figure A39.** Gen-FTP deployment can be accomplished by utilizing the energy available when OLSR is in its idle mode.

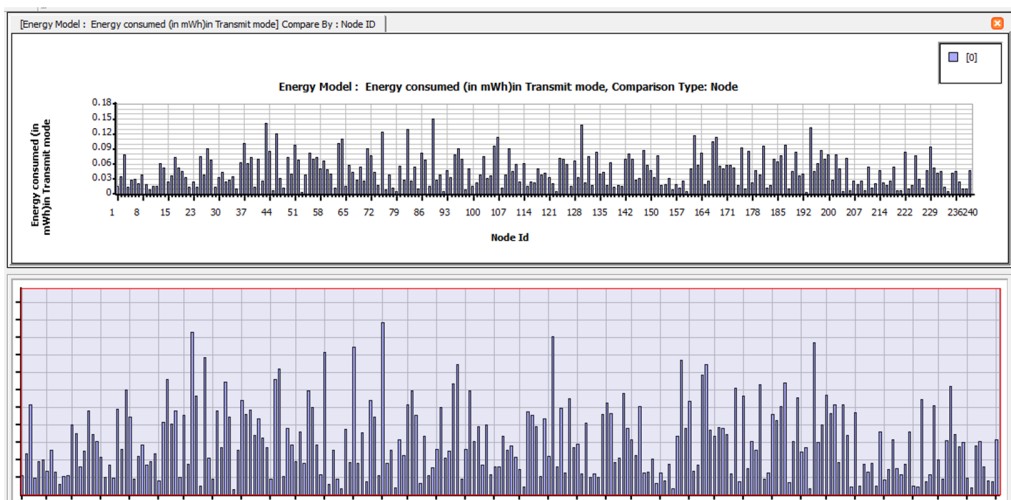

**Figure A40.** Gen-FTP deployment can be accomplished by utilizing the energy available when OLSR is in its Transmit mode.

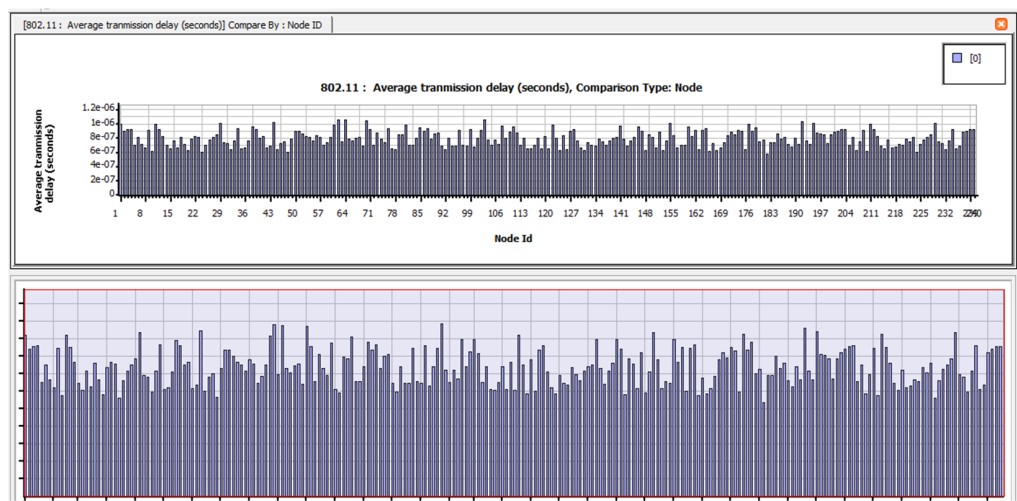

**Figure A41.** The average time delay required for transmissions during a Gen-FTP S-frame deployment of LAR1.

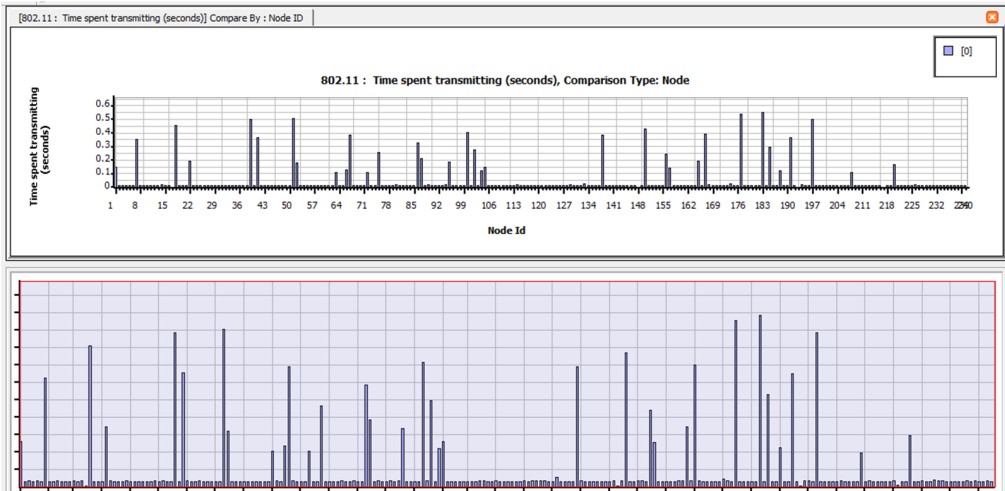

**Figure A42.** Transmission time during the deployment of the Gen-FTP to the LAR1.

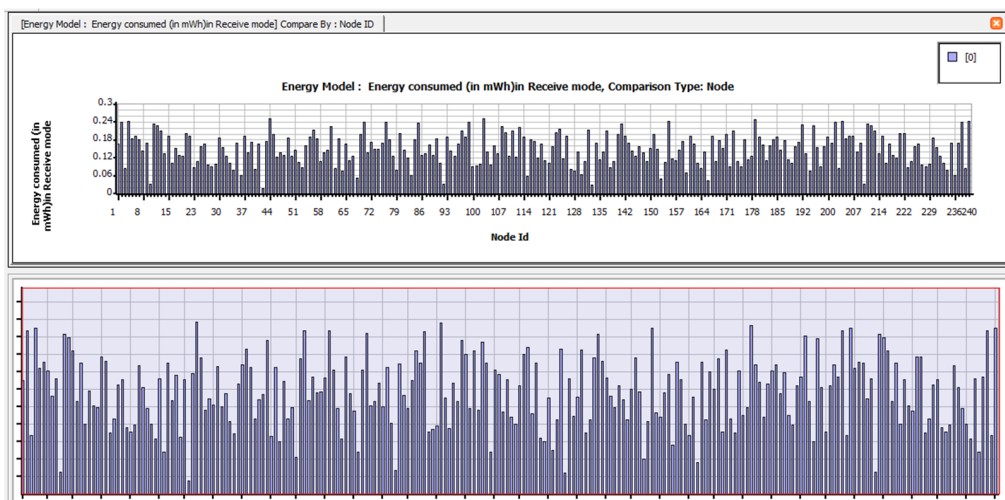

**Figure A43.** Gen-FTP deployment can be accomplished by utilizing the energy available when LAR1 is in its Receive mode.

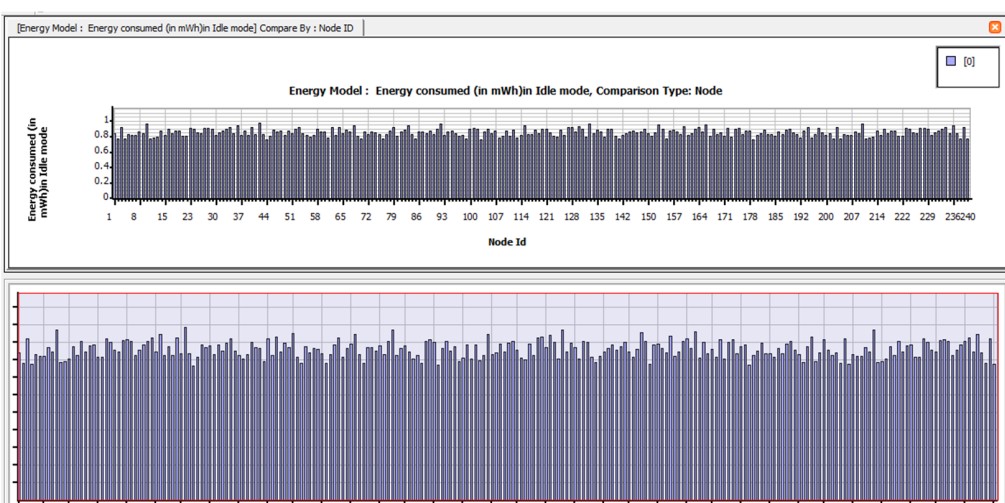

**Figure A44.** Gen-FTP deployment can be accomplished by utilizing the energy available when LAR1 is in its idle mode.

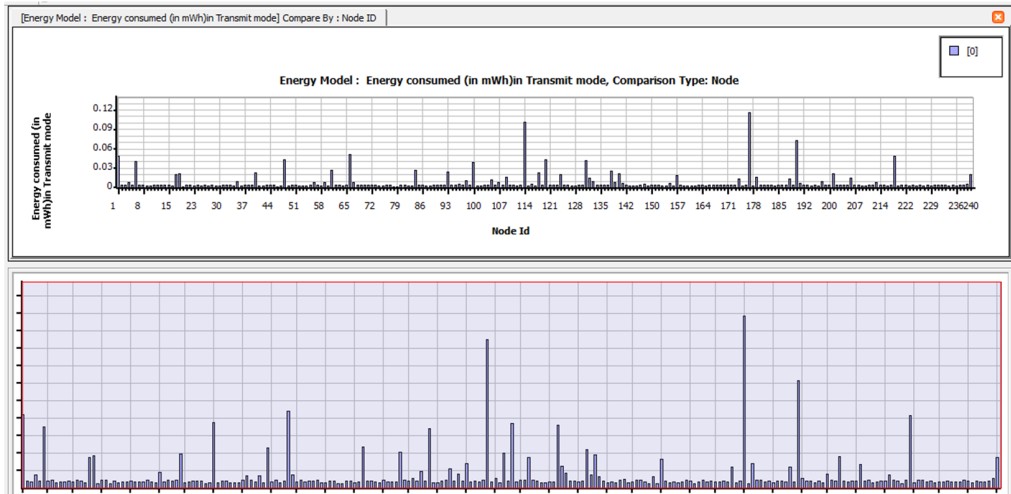

**Figure A45.** Gen-FTP deployment can be accomplished by utilizing the energy available when LAR1 is in its Transmit mode.

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
