# Peer review of "Reliable Data Transmission in Underwater Wireless Sensor Networks Using a Cluster-Based Routing Protocol Endorsed by Member Nodes"

_electronics, doi:10.3390/electronics12061287_

Round 1

Reviewer 1 Report

Figures 6-50 include in Appendix Pages. 

Explain for those figure are in single line. Explain More technical interpretation, comparative, reasons and etc. 

Improve the Figures Quality as per Standard .

The conclusion and abstract are to be technically improved. 

Include statistical Table of information for all the Graphs. (include More tables)

Grammar and repair the some of sentences of the manuscript.  

Author Response

Thank you for reviewing our manuscript, sir.

  1. As per the instructions, we have included in Appendix pages (Figures 6 – 50).
  2. As per the suggestions, we have thoroughly revised accordingly.
  3. As per the requirements, the quality of the figures is improved for visibility.
  4. As per the suggestions, we have thoroughly revised the conclusion and abstract accordingly.
  5. As per the instructions, we have incorporated the statistical tables for all the graphs.
  6. Our thanks to those who provided feedback, which led to the addition of the highlighted discussions of the manuscript and carefully revised the manuscript with proper grammar.

Reviewer 2 Report

Few Snchoronization/Abbreviations issues, i.e., in the abstract, underwater acoustic sensor networks were abbreviated as UASNs while Member Nodes supported Cluster Based Routing Protocol abbreviated as MNC- BRP. Please consider case sensitivity. 

Also, abbreviate terms on their first occurrence. 

Please elaborate on the first sentence of the introduction and try to differentiate between underwater acoustic sensor networks and IoUT.

Is there any difference between UWSN and UASN? if you are using these terms alternatively, then please consider a footnote to elaborate. 

The authors look confused between UWSN, and UAWSN, while UWSN is also used for United States Navy Wireless Network too.

What are your proposed work's limitations? Also, consider writing its scope clearly. 

Is there any tradeoff that occurred while getting your objectives? Consider making a section for this.

Results need to be elaborate. 

Author Response

Thank you for reviewing our manuscript, sir.

  1. We thank the reviewer comment. We have defined all abbreviations when we are quoting first time, following reviewer suggestion.
  2. As per the suggestions, we have elaborated the introduction part thoroughly and revised accordingly.
  3. As per the requirements, we have commonly interpreted as UAWSN.
  4. As per the suggestions, we have thoroughly cleared the terms accordingly (UWSN, UAWSN).
  5. In present work, we have limited only for power and delay parameters and in future we will extend it for other parameters like BER, throughput etc.
  6. For the Telnet deployment application, LAR1 spends 11 msec for the Telnet deployment application
  7. Our thanks to those who provided feedback, which led to the addition of the highlighted discussions of the manuscript and carefully revised the manuscript with the elaboration of results.

Reviewer 3 Report

The authors addressed an interesting and topical topic. They submitted an intense work obtaining valuable results.

It would have been useful to try a comparison of the overall characteristics of the proposed underwater network with the already implemented and functional networks. In this way, advantages and disadvantages would have appeared more clearly in relation to real cases.

Author Response

Thank you for reviewing our manuscript, sir.

Our thanks to those who provided feedback, which led to the addition of the highlighted discussions of the manuscript and carefully revised the manuscript with the comparison of the overall characteristics of the proposed underwater network with the already implemented and functional networks.

Reviewer 4 Report

In this paper, the authors proposed a study a reliable data transmission in underwater wireless sensor networks using a cluster-based routing protocol. However, I have some comments.

1.       Please perform thorough revision to remove any grammatical errors.

2.       Literature review is incomplete. Latest works in technical literature which study ultra-reliable and low latency communications and NOMA in futuristic networks should be added such as

[a] Ranjha, A., Kaddoum, G., Rahim, M. and Dev, K., 2022. URLLC in UAV-enabled multicasting systems: A dual time and energy minimization problem using UAV speed, altitude and beamwidth. Computer Communications187, pp.125-133.

[b] Narsani, H.K.., Dev, K., Memon, F.H. and Qureshi, N.M.F., 2022. Leveraging UAV-assisted communications to improve secrecy for URLLC in 6G systems. Digital Communications and Networks.

[c] Asif, M., Ihsan, A., Khan, W.U., Ranjha, A., Zhang, S. and Wu, S.X., 2022. Energy-Efficient Backscatter-Assisted Coded Cooperative-NOMA for B5G Wireless Communications. IEEE Transactions on Green Communications and Networking.

3.       The authors should add pseudo-code and time-complexity of the proposed MNCBRP algorithm.

4.       The authors should comment on the convergence of MNCBRP algorithm. 

Author Response

Thank you for reviewing our manuscript, sir.

Our thanks to those who provided feedback, which led to the addition of the highlighted discussions of the manuscript and carefully revised the manuscript with the comparison of the overall characteristics of the proposed underwater network with the already implemented and functional networks.

  1. We thank the reviewer comment. We have thoroughly proofread the entire manuscript and made the language is clear, following reviewer suggestion.
  2. As per the instructions, we have included in the references i.e., [48, 49 and 50].
  3. As per the requirement, we have added pseudo code which is highlighted.
  4. Our thanks to those who provided feedback, which led to the addition of the highlighted discussions. Our research focus is primarily on the parameters power and delay but not for convergence, but it may be considered in our future work.

Reviewer 5 Report

It is an interesting work on the analysis of the protocol that optimizes the resources of wireless sensor networks.

The modeling of the network and the implementation of the network protocol assembly of the simulation seems sound. However, the presentation of results is confusing and does not give a clear picture of the performance of the network. It would be better to present the results in a way that is clearer and more meaningful.

The manuscript contains too many graphs on the performance of different network parameters, so it would be advisable to reduce the number of illustrations presented, as there are too many (Figure 6 to 50 in particular).

Author Response

Thank you for reviewing our manuscript, sir.

  1. Our thanks to those who provided feedback, which led to the addition of the highlighted discussions of the manuscript and carefully revised the manuscript with the comparison of the overall characteristics of the proposed underwater network with the already implemented and functional networks and presented the results with meaning full.
  2. As per the instructions, we have included in Appendix pages (Figures 6 – 50).

Reviewer 6 Report

Relatively to the paper titled “Reliable Data Transmission in Underwater Wireless Sensor Networks Using a Cluster-Based Routing Protocol Endorsed by Member Nodes” (paper ID: Electronics-2203918), I believe that the authors have done an overall good job in terms of quality and originality; the paper is well-written and organized. The English used is sufficiently readable. However, some portions of the manuscript should be furtherly deepened and more detailed. Also, other issues in the manuscript should be solved to improve the article’s readability.

-       The authors should review the abstract to better introduce the main obtained results (also reporting numerical data) to strengthen the paper’s content.

-       Relatively to the Introduction, the authors are suggested to better introduce the carried out work and research activity, as well as the novelties of the presented UWSN routing protocol compared to other ones already reported in the scientific literature.

-       The acronyms should be clarified at the first appearance (Gen-FTP, CBR, BS, etc.).

-       Figure 3 is unclear; the authors should revise them. Perhaps the authors could write the references to the nodes in white colour.

-       Relatively to Figures 6 - 50, the authors are suggested to improve their legibility and quality; besides, the authors should better introduce these figures with a few lines of comment without reporting merely a list of 44 figures. Alternatively, the authors could move the simulation graph into the supporting materials, leaving the following summarizing graphs and tables in the paper.

-       The authors should review the reference citation in the bibliography to comply with the journal template (https://www.mdpi.com/journal/electronics/instructions).

Author Response

Thank you for reviewing our manuscript, sir.

Our thanks to those who provided feedback, which led to the addition of the highlighted discussions of the manuscript and carefully revised the manuscript with the comparison of the overall characteristics of the proposed underwater network with the already implemented and functional networks.

  1. As per the suggestions, the abstract part is changed with meaning to strengthen the paper’s content.
  2. As per the instructions, we have included in the references.
  3. We thank the reviewer comment. We have defined all abbreviations when we are quoting first time, following reviewer suggestion.
  4. We thank the reviewer comment. We have resized and improved Figures 3, following reviewer suggestion.
  5. As per the instructions, we have included in Appendix pages (Figures 6 – 50).
  6. Our thanks to those who provided feedback, which led to the reference citation in the bibliography to comply with the journal template.

We strongly feel that the review is extremely good.

Once again we are thanking you for your comments and suggestions.

Round 2

Reviewer 1 Report

Accepted as it is..

Author Response

Thank you for reviewing our manuscript, sir.

Reviewer 2 Report

There are some abbreviations that still need to be told at their first occurrence. I.e., AUVs and ROVs on line 75. Some other abbreviations are used before abbreviating a word, i.e., LEECH, GAF, HEED, etc. 

Also, some abbreviations are used only once at their first occurrence only; i.e., GPS, etc. 

Results need to be elaborate. E.g., Why your proposed solution performs better than others at any specific point. I.e., Fig 52

Author Response

Thank you for reviewing our manuscript, sir.

  1. We thank the reviewer comment. As the Reviewer commented, the abbreviations have been expanded on their first occurrence. In addition, we have additionally added the List of acronyms in the manuscript Page number
  2. As per the suggestions, we have thoroughly cleared the abbreviations i.e., GPS.
  3. As per the suggestions, we have thoroughly elaborated the results accordingly. Till now the authors are worked with the generic architecture we have considered Random Waypoint Mobility.

Reviewer 4 Report

All the comments have been adequately addressed.

Author Response

(The authors gave the same response as above.)

Reviewer 5 Report

Improve figure 3b as much as possible.

You must indicate the numbering of the manuscript consecutively and use another numbering in the figures in the appendix. (As an example, figure 51 would become figure 6).

Define the acronyms for the LARI and OSLR protocols before using them.

His work proposes a new network architecture, however it is not benchmarked against other architectures.

Author Response

Thank you for reviewing our manuscript, sir.

  1. As per the requirements, the quality of figures 3a & 3b is improved for visibility.
  2. As per the instructions, we have included the consecutive numbering of Figures in both contexts (Figure 1 to Figure 10) and in the Appendix (Figure 11 to Figure 55).
  3. We thank the reviewer comment. We have defined all abbreviations when we are quoting first time, following reviewer suggestion.
  4. Our thanks to those who provided feedback, which led to the addition of the highlighted discussions of the manuscript and carefully revised the manuscript. As of now the authors are implemented in conventional architecture with deploying applications as Constant Bit Rate (CBR), we have considered real time applications such as Teletype network (Telnet) protocol, Supervisory- frame (S-frame) and Generic –File Transfer Protocol (Gen-FTP) .

Round 3

Reviewer 5 Report

I consider that the requested changes were made and the manuscript can be published.